# Major nuclear locales define nuclear genome organization and function beyond A and B compartments

Omid Gholamalamdari[1†], Tom van Schaik[2†], Yuchuan Wang[3†], Pradeep Kumar[1†], Liguo Zhang[1†], Yang Zhang[3†], Gabriela A Hernandez Gonzalez[1], Athanasios E Vouzas[4], Peiyao A Zhao[4], David M Gilbert[4*], Jian Ma[3*], Bas van Steensel[2*], Andrew S Belmont[1,5,6*]

[1]Department of Cell and Developmental Biology, University of Illinois at Urbana-Champaign, Urbana, United States; [2]Division of Gene Regulation and Oncode Institute, Netherlands Cancer Institute, Amsterdam, Netherlands; [3]Ray and Stephanie Lane Computational Biology Department, School of Computer Science, Carnegie Mellon University, Pittsburgh, United States; [4]San Diego Biomedical Research Institute, San Diego, United States; [5]Center for Biophysics and Quantitative Biology, University of Illinois at Urbana-Champaign, Urbana, United States; [6]Carl R. Woese Institute for Genomic Biology, University of Illinois at Urbana-Champaign, Urbana, United States

*For correspondence:
gilbert@sdbri.org (DMG);
jianma@cs.cmu.edu (JM);
b.v.steensel@nki.nl (BvS);
asbel@illinois.edu (ASB)

†These authors contributed equally to this work

Competing interest: The authors declare that no competing interests exist.

## eLife Assessment

In this **valuable** study, the authors integrate several datasets to describe how the genome interacts with nuclear bodies across distinct cell types and in Lamin A and LBR knockout cells. They provide **convincing** evidence to support their claims and particularly find that specific genomic regions segregate relative to the equatorial plane of the cell when considering their interaction with various nuclear bodies.

**Abstract** Models of nuclear genome organization often propose a binary division into active versus inactive compartments yet typically overlook nuclear bodies. Here, we integrated analysis of sequencing and image-based data to compare genome organization in four human cell types relative to three different nuclear locales: the nuclear lamina, nuclear speckles, and nucleoli. Although gene expression correlates mostly with nuclear speckle proximity, DNA replication timing correlates with proximity to multiple nuclear locales. Speckle attachment regions emerge as DNA replication initiation zones whose replication timing and gene composition vary with their attachment frequency. Most facultative LADs retain a partially repressed state as iLADs, despite their positioning in the nuclear interior. Knock out of two lamina proteins, Lamin A and LBR, causes a shift of H3K9me3-enriched LADs from lamina to nucleolus, and a reciprocal relocation of H3K27me3-enriched partially repressed iLADs from nucleolus to lamina. Thus, these partially repressed iLADs appear to compete with LADs for nuclear lamina attachment with consequences for replication timing. The nuclear organization in adherent cells is polarized with nuclear bodies and genomic regions segregating both radially and relative to the equatorial plane. Together, our results underscore the importance of considering genome organization relative to nuclear locales for a more complete understanding of the spatial and functional organization of the human genome.

## Introduction

Over 100 years of cytology has recognized characteristic features of nuclear genome organization, including blocks of condensed chromatin enriched at the nuclear and nucleolar peripheries and varying degrees of condensation/decondensation throughout the nuclear interior. Foci of active transcription are concentrated at the edges of chromosome territories and sub-territory condensed chromatin masses and depleted at the nuclear periphery (*Belmont, 2022*; *Bernhard and Granbould, 1963*; *Cremer et al., 2015*; *Cremer and Cremer, 2006*; *Misteli, 2020*; *Politz et al., 2013*; *Politz et al., 2016*; *Takizawa et al., 2008*; *van Steensel and Belmont, 2017*). Two current models for nuclear genome organization prevail (*Belmont, 2022*).

The radial genome organization model describes a radial gradient of higher gene expression activity toward the nuclear center (*Bickmore, 2013*; *Croft et al., 1999*; *Girelli et al., 2020*; *Kölbl et al., 2012*; *Küpper et al., 2007*; *Takizawa et al., 2008*). The binary model of genome organization describes the differential intranuclear positioning of two discrete chromatin states: heterochromatin primarily to the nuclear lamina, and to a lesser extent the nucleolar periphery and clusters of PCH, and euchromatin to the remaining nuclear space or 'interior'. Originally supported by the differential intranuclear segregation of Alu repeat-enriched, early replicating versus LINE1 repeat-enriched, late replicating chromosome bands (*Korenberg and Rykowski, 1988*; *Bolzer et al., 2005*; *Solovei et al., 2009*), three orthogonal genomic methods – DamID, Repli-seq, and Hi-C – have segmented the genome into similar Lamin Associated Domains (LADs)/ late replicating/ B compartment versus inter-LAD (iLADs), early replicating/ A compartment binary divisions (*Guelen et al., 2008*; *Kind et al., 2013*; *Lieberman-Aiden et al., 2009*; *Peric-Hupkes et al., 2010*; *Ryba et al., 2010*; *White et al., 2004*).

However, neither model acknowledges the considerable cell-type-specific variability in nuclear shape and size, as well as the size, number, and relative positioning of nuclear bodies. This variability could dramatically affect chromosome trajectories and the distribution of active and inactive chromatin regions among nuclear locales. Additionally, neither model acknowledges the variation between cell types in how different heterochromatin regions distribute between the nuclear periphery, the nucleolar periphery, and the nuclear interior as well as how active genes might segregate to different regions of the nuclear interior. In fact, these differences comprise a major element of histology and pathology textbooks and are used routinely for clinical diagnosis (*Fischer, 2020*; *Skinner and Johnson, 2017*).

Indeed, high-resolution Hi-C has identified two active A subcompartments and three major repressive B subcompartments (*Rao et al., 2014*). Similarly, TSA-seq, NAD-seq, and SPRITE have distinguished between types of heterochromatin localizing preferentially at nucleoli versus the nuclear lamina and have revealed a type of active chromatin specifically localizing near nuclear speckles (*Chen et al., 2018*; *Németh et al., 2010*; *Quinodoz et al., 2021*; *Quinodoz et al., 2018*; *van Koningsbruggen et al., 2010*; *Vertii et al., 2019*; *Zhang et al., 2021*). However, a major limitation with these previous studies is that they typically focused on one type of measurement (i.e. Hi-C, DamID, or NAD-seq), one or two nuclear compartments, and/or one or a limited number of cell types.

To address these problems, we integrated multiple spatial and functional genomic measurements and light microscopy across multiple cell lines. We focused on the specific goal of relating nuclear genome features to three nuclear locales – the nuclear periphery (NP), nucleoli, and nuclear speckles – while also developing improvements to DamID (*van Schaik et al., 2020*), TSA-seq (*Zhang et al., 2021*), and Repli-seq (*Zhao et al., 2020*). In a companion paper, we extended and validated our improved TSA-seq 2.0 protocol to map the nuclear genome relative to the nuclear lamina and nucleoli (*Kumar et al., 2024*), adding to our previous TSA-seq 2.0 mapping relative to nuclear speckles (*Zhang et al., 2021*) and our improved-resolution 16-fraction Repli-seq (*Zhao et al., 2020*).

Here, we integrate this nuclear locale genomic mapping with light microscopy imaging of the same three nuclear locales to investigate how the genome is both spatially and functionally organized relative to these nuclear locales across four human cell types- hTERT-immortalized human foreskin fibroblasts (HFFc6, abbreviated hereafter as HFF), H1 human embryonic stem cells (hESCs), HCT116 (HCT) colon carcinoma epithelial cells, and K562 erythroleukemia cells.

Our results show that both radial and/or binary divisions of the genome are oversimplified. Multiple types of active versus inactive chromatin states can be recognized by their varying association with nuclear locales. Moreover, nuclear genome organization varies among cell types both because of changes in genome positioning relative to individual nuclear locales as well as changes in the

intranuclear positioning of these locales relative to each other. Gene expression shows much stronger correlation with distance to nuclear speckles versus either the nuclear lamina or nucleoli. Finally, we demonstrate a nuclear polarity to the genome organization beyond a simple radial axis which further spatially divides active versus repressive genomic states as well as DNA replication timing.

## Results

### Cell types differ widely in morphology and arrangement of nuclear locales

Prior genomic analyses of nuclear locales have largely ignored variation in nuclear morphology. However, differences in position relative to nuclear locales can result from either differences in chromatin location or to differences in the relative positioning or even morphologies of nuclear locales. Therefore, we began by comparing nuclear, nucleolar, and nuclear speckle sizes and shapes, as well as nucleolar and nuclear speckle positioning relative to each other and the NP, across the four human cell lines – H1, HCT116, HFF, and K562.

To measure sizes and shapes of nuclei and nuclear bodies, cell lines were stained for nuclear speckles (hereafter referred to simply as speckles), nucleoli, and the NP with anti-SON, anti-MKI67IP, and anti-glycosylated nucleoporin (anti-RL1) antibodies, respectively, and for DNA using DAPI. Deconvolved wide-field microscopy images (*Figure 1A*) were then segmented to generate nuclear 3D models (*Figure 1B*, *Figure 1—figure supplement 1A*).

K562 and H1 nuclei have larger volume (V), lower surface area (SA), and are rounder (lower SA/V) as compared to HCT and HFF nuclei (*Figure 1C*). Although K562 and H1 speckles distribute radially in 3D, surrounding nucleoli and concentrating within the nuclear interior, HCT and HFF speckles preferentially distribute in the equatorial (center) z-planes while being depleted radially near the NP within these equatorial planes (*Figure 1B*).

However, other nuclear morphology features did not segregate with flat versus round nuclei. K562 nuclei are lobulated and have deep invaginations of the nuclear lamina (*Figure 1A*). Individual nuclear speckles are significantly larger in HCT116 cells, even though total nuclear speckle volumes are similar across all four cell lines (*Figure 1D*). H1 and HCT116 have large, often single, nuclear-centrally located nucleoli, while HFF and K562 have smaller, multiple nucleoli distributed between speckles (*Figure 1A, B and D*). Total nucleolar volumes are largest in HCT116 and lowest in HFF (*Figure 1D*).

We compared relative arrangements of these nuclear locales across cell lines by measuring the nearest distances between these locales (*Figure 1—figure supplement 1B*). Rounder nuclei (K562, H1) have significantly larger speckle-to-NP and nucleolus-to-NP distances compared to flatter nuclei (HFF, HCT116) as expected (*Figure 1—figure supplement 1C*, *Figure 1E*). But speckle-to-nucleoli distances in K562 are lower than other cell lines, with HFF showing noticeably larger speckle-to-nucleoli distances (*Figure 1—figure supplement 1C*, *Figure 1E*). While total speckle and nucleolar volumes scale linearly with nuclear volume in all cell lines, curiously total speckle volume correlates more strongly with nuclear surface area versus volume (*Figure 1—figure supplement 1D*).

Overall, we found a large variation in the sizes and shapes of nucleoli and nuclear speckles as well as their spatial arrangement relative to each other and to the NP. However, despite various cell-line-specific differences in nuclear locale morphologies, the four cell lines divide approximately into round (K562, H1) nuclei with radial organization of nuclear bodies versus flat nuclei (HCT116, HFF) with nuclear bodies distributed differentially relative to distance from both the x-y equatorial plane as well as distance from the nuclear center. This division is further supported by Principal Component Analysis (PCA) using 14 morphological measures (*Figure 1F*, *Figure 1—figure supplement 1F*).

### Global differences in nuclear genome organization in different cell types revealed by DamID and TSA-seq

We next surveyed genome organization relative to these nuclear locales using a combination of DamID and TSA-seq. DamID is a well-established molecular proximity assay; DamID applied to the nuclear lamina divides the genome into lamina-associated domains (LADs) versus non-associated 'inter-LADs' or 'iLADs' (*Guelen et al., 2008*; *van Steensel and Belmont, 2017*). In contrast, TSA-seq measures relative distances to targets on a microscopic scale corresponding to 100 s of nm to ~1 µm based on the measured diffusion radius of tyramide-biotin free-radicals (*Chen et al., 2018*). Exploiting the

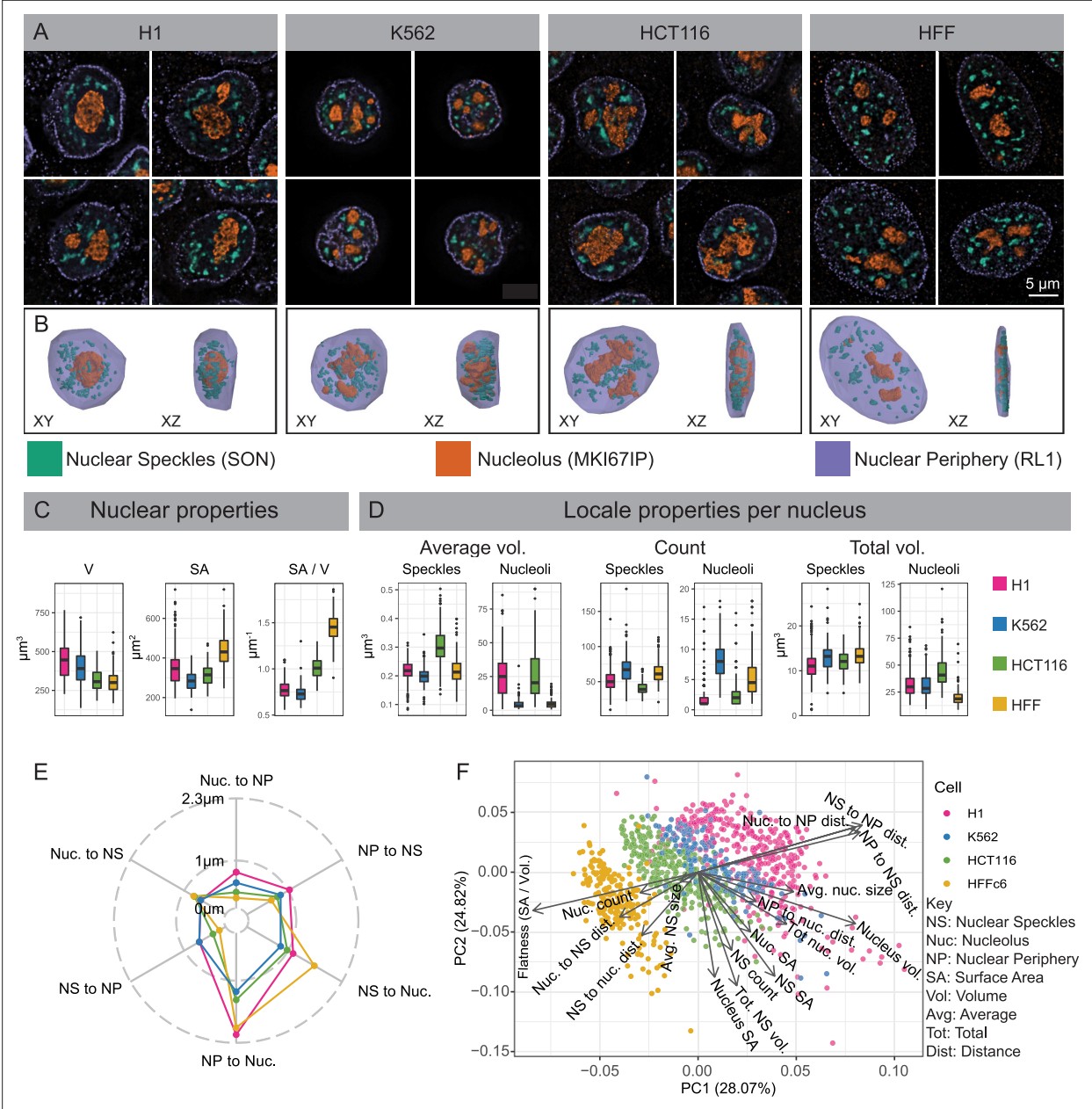

**Figure 1.** Differences in nuclear and nuclear body morphology and relative positioning among four cell types. (**A–B**) Wide-field deconvolution light microscopy (**A**) and 3D solid models in XY and XZ orientations (**B**). Immunostaining of nuclear periphery (NP), nucleoli, and nuclear speckles using antibodies against RL1 (nuclear pore, purple), MKI67IP (nucleolar GC, orange), and SON (nuclear speckle, green) in H1, K562, HCT116, and HFF cells. (**C**) Comparing Volume (V), surface area (SA), and roundness (SA/V) of nuclei measured from NP 3D solid models in H1 (red), K562 (blue), HCT116 (green), and HFF (yellow) cells. (**D**) Comparison across cell lines of nucleolar and nuclear speckle (NS) numbers, average volumes, and summed volumes per nucleus. Individual NS are significantly larger in HCT116 cells, even though total NS volumes are similar across all four cell lines. (**E**) Pairwise average distances between locales (asymmetric, as defined in *Figure 1—figure supplement 1B*). (**F**) Principal Component (PC) 1 (x-axis) versus PC2 (y-axis) scatterplot using PCA of 14 morphological features reveals adjacent clustering of H1 and K562 cells with HCT116 and HFF clusters closer to each than to H1 and K562 cells. Each scatterplot point represents an individual cell. Arrow lengths show the quality of representation and arrow direction show the loadings on PC1 and PC2.

The online version of this article includes the following figure supplement(s) for figure 1:

**Figure supplement 1.** Comparisons of nuclear and nuclear body morphologies and the intranuclear relative positioning of nuclear bodies across four cell types.

measured exponential decay of the tyramide-biotin free-radical concentration, we showed how the mean distance of chromosomes to nuclear speckles could be estimated from the TSA-seq data to an accuracy of ~50 nm, as validated by FISH (*Chen et al., 2018*). While lamin DamID segments LADs most accurately, lamin TSA-seq provides distance information not provided by DamID- for example, variations in relative distances to the nuclear lamina of different iLADs and iLAD regions. These differences between the lamin DamID and TSA-seq signals are also crucial to a computational approach, SPIN, that segments the genome into multiple states based on their varying nuclear localization, including biochemically and functionally distinct lamina-associated versus near-lamina states (*Dekker et al., 2024*; *Wang et al., 2021*).

Thus, lamin DamID and TSA-seq complement each other, providing more information together than either one separately. For these reasons, we set out to develop both SON and nucleolar DamID and nucleolar TSA-seq with the goal of mapping the genome relative to the nuclear lamina, nuclear speckles, and nucleoli in each case using both DamID and TSA-seq. We successfully developed nucleolar TSA-seq, which we extensively validated using comparisons with two different orthogonal genome-wide approaches (*Kumar et al., 2024*)- NAD-seq, based on the biochemical isolation of nucleoli, and previously published direct microscopic measurements using highly multiplexed immuno-FISH (*Su et al., 2020*). As previously demonstrated for both SON and lamin TSA-seq (*Chen et al., 2018*), nucleolar TSA-seq was also robust in the sense that multiple target proteins showing similar nucleolar staining showed similar TSA-seq results (*Kumar et al., 2024*); this robustness is intrinsic to TSA-seq being a microscopic rather than molecular proximity assay, and therefore not sensitive to the exact molecular binding partners and molecular distance of the target proteins to the DNA.

In contrast, we were unsuccessful with both the SON and nucleolar DamID. SON DamID showed narrow, several hundred bp peaks near the promoters of most active genes, similar to previous SON ChIP-seq profiles (*Kim et al., 2016*), consistent with measurement of the local accumulation of SON molecules at these sites rather than chromosome contact with nuclear speckles. Nucleolar DamID instead showed broad positive peaks over large chromatin domains, largely overlapping with LADs mapped by LMNB1 DamID (*Wang et al., 2021*). However, this nucleolar DamID signal, while strongly correlated with lamin DamID, showed poor correlation with either NAD-seq or nucleolar distances mapped by multiplexed immuno-FISH (*Kumar et al., 2024*). We suspect the problem is that DamID is a molecular rather than microscopic proximity assay; therefore, both the SON and nucleolar DamID output signals should be disproportionally dominated by the small fraction of target proteins juxtaposed in sufficient proximity to the DNA to produce a signal, rather than by the amount of protein concentrated in the microscopically adjacent target nuclear body.

For these reasons, our analyses focused on those measurements that proved robust in measuring relative proximity to the nuclear lamina, nucleoli, and nuclear speckles locales: lamina (LMNB1) DamID and, as described in previous papers, lamina (LMNB1) and nucleolar (MKI67IP; *Kumar et al., 2024*), and speckle (SON; *Zhang et al., 2021*) TSA-seq 2.0.

Speckle TSA-seq is highly conserved across all four cell types (*Figure 2A*), as described previously (*Zhang et al., 2021*) and supported here by the high correlations for speckle TSA-seq comparisons between pairs of cell lines, despite a lower dynamic range in HCT116 and especially HFF (*Figure 2B*).

In contrast, genome-wide correlations in pair-wise cell line comparisons were substantially lower for lamina DamID and for both lamina and nucleolus TSA-seq (*Figure 2A–B*, *Figure 2—figure supplement 1A–C*). For example, LADs near the ends of long chromosome arms switch from a strong lamina association in HFF and HCT116 to a more interior and nucleolar association in H1 and K562 (*Figure 2—figure supplement 1A*). While in HCT116 cells local maxima in nucleolar TSA-seq align with LADs, in HFF, H1, and K562 cells small peak and 'peak-within-valley' nucleolar TSA-seq local maxima instead align with speckle TSA-seq peaks (*Figure 2—figure supplement 1B*).

Thus, whereas genome positioning relative to speckles is highly conserved across cell lines, despite variations in speckle intranuclear positioning (*Figure 1*), genome positioning relative to the lamina and nucleoli shows major variations across different cell types.

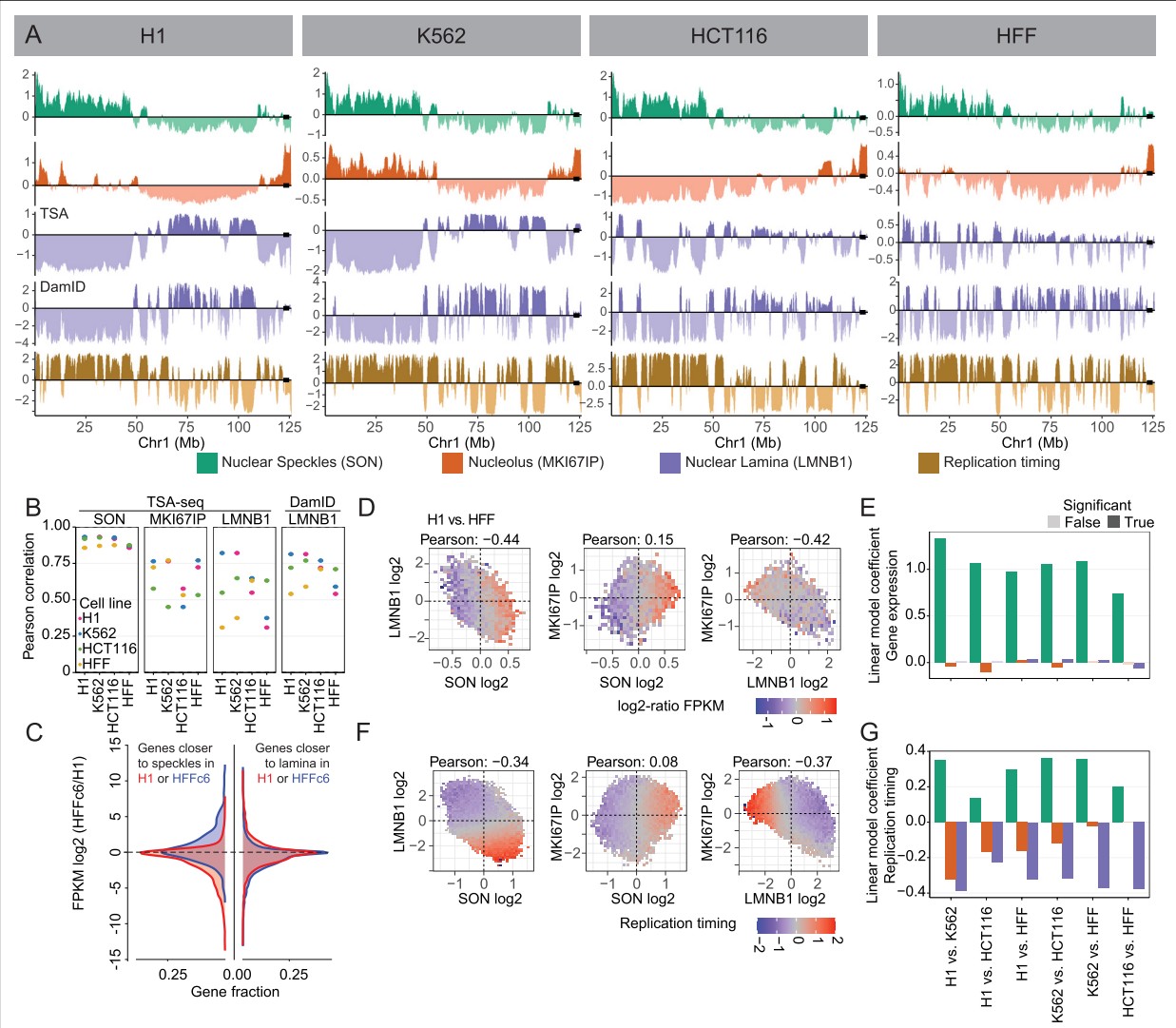

**Figure 2.** Global changes in nuclear genome intranuclear positioning and their correlations with changes in gene expression and DNA replication timing. (**A**) Chr1 left arm browser view suggests chromosome trajectory alternating between positioning of early replicating regions near nuclear speckles and late-replicating regions at either the nucleolar (H1, K562) or the nuclear periphery (HCT116, HFF); Top to bottom- nuclear speckle (SON, green), nucleolus (MKI67IP, orange), and nuclear lamina (LMNB1, purple) TSA-seq, LMNB1 (purple) DamID, and 2-fraction Repli-seq (Replication timing, brown). Left to right- H1, K562, HCT116, and HFF cells. (**B**) Comparison of Pearson correlations of TSA-seq and DamID datasets between different cell lines (H1, K562, HCT116, HFF) reveals much higher conservation of genome positioning relative to nuclear speckles, which is largely conserved, versus nucleoli or nuclear lamina; (**C**) Left- Genes positioned closer (blue)/ further (red) to nuclear speckles show a bias toward increased / decreased gene expression in HFF versus H1 cells. Fraction of genes showing significant difference in relative positioning to nuclear speckles (gene fraction, x-axis) versus log2 (HFF FPKM / H1 FPKM) (y-axis); Right- Similar comparison for genes positioned closer (blue) / further (red) to nuclear lamina does not show such bias; (**D–E**) Changes in gene expression between H1 and HFF cells vary largely as function of changes in nuclear speckle rather than nuclear periphery or nucleolar relative positioning; (**D**) 2D histograms showing mean ratio changes in gene expression between H1 versus HFF cells (log2 ratios of FPKM, color-coded) for binned genes as function of their changes in z-normalized TSA-seq scores (left to right: LMNB1 (y) vs SON (x), MKI67IP (y) vs SON (x), MKI67IP (y) vs LMNB1 (x)); (**E**) Linear modeling of changes in gene expression versus z-normalized TSA-seq score changes reveals significantly larger coefficients (dependence) for SON (green) versus MKI67IP (orange) or LMNB1 (blue). (**F–G**) Similar comparison as in (**D–E**), but for replication timing (2-fraction Repli-seq), shows changes in DNA replication timing are function of both changes in SON and LMNB1 TSA-seq.

The online version of this article includes the following figure supplement(s) for figure 2:

**Figure supplement 1.** Global changes in nuclear genome intranuclear positioning and their correlations with changes in gene expression and DNA replication timing.

## Differences in gene expression primarily correlate with differences in genome positioning relative to speckles, whereas differences in DNA replication timing correlate with differences in relative distances to all three nuclear locales

Given both the binary and radial models of nuclear genome organization, there has been a long-standing expectation that changes in gene expression would correlate with changes in distance relative to the NP. More recently, the strong correlation of gene expression levels with position along a nuclear lamina – nuclear speckle axis in K562 cells, as measured by TSA-seq, led to the suggestion that distance along this axis was a more directly functionally relevant metric with respect to gene expression (*Chen et al., 2018*).

The large variations in genome positioning relative to the lamina and nucleoli among the four cell types, described in the previous section, now allowed us to ask how exactly differences in gene expression and/or DNA replication timing correlate with differences in distances to each of the different nuclear locales.

We had previously identified genomic regions 100 kb or larger, comprising ~10% of the genome, that showed statistically significant differences in speckle TSA-seq smoothed scores in pairwise comparisons between the same four cell lines (*Zhang et al., 2021*). Genes within these regions showed a strong positive correlation between differences in their gene expression and speckle positioning. Chromosome regions with different position relative to speckles typically show inverse differences in their position relative to the lamina (*Figure 2—figure supplement 1D*, top panels), although individual genome regions have variable magnitude and even direction of these differences (*Figure 2—figure supplement 1E*). However, the reverse is not true. Pairwise cell-line comparisons reveal ~40–70% of the genome shows statistically significant differences in lamina TSA-seq over regions 100 kb or larger, with most of these regions showing little or no differences in speckle TSA-seq scores (*Figure 2—figure supplement 1D–E*, bottom panels).

Genes within genomic regions displaying significant differences in speckle TSA-seq in cell line comparisons tend to show increased expression when they are closer to speckles and decreased expression when they are further from speckles (*Figure 2C*), as described previously (*Zhang et al., 2021*). In contrast, genes within genomic regions which in pair-wise comparisons of cell lines show a statistically significant difference in lamina TSA-seq show no obvious trend in their expression differences (*Figure 2C*).

Binning all genes as a function of differences in their normalized speckle, nucleolar, and lamina TSA-seq scores show a positive correlation of differences in gene expression with changes in speckle TSA-seq scores but no apparent correlation with differences in lamina or nucleolar TSA-seq scores (*Figure 2D*). Linear modeling for correlation between changes in gene expression and changes in TSA-seq reveals a substantially stronger correlation for positioning relative to speckles, as compared to positioning relative to the lamina or nucleoli (*Figure 2E*, *Figure 2—figure supplement 1F*); this pattern holds genome-wide (*Figure 2E*) as well as for correlations over specific genome region types (LAD, iLAD, speckle TSA-seq maxima; *Figure 2—figure supplement 1F*). Instead, pair-wise cell type comparisons revealed significant correlations between changes in DNA replication timing and changes in TSA-seq with respect to all three locales – speckle, lamina, and nucleoli (*Figure 2F–G*).

In summary, whereas gene expression primarily correlates with relative distance to nuclear speckles, DNA replication timing correlates with relative distance to all three nuclear locales, despite the variable positioning of nuclear speckles and nucleoli across cell types.

## Both Type I and Type II speckle attachment regions are gene expression hot-zones and DNA replication IZs but show varying gene composition and DNA replication timing

Previously, we showed that SON TSA-seq local maxima align with gene expression 'hot-zones' embedded within larger iLAD/A compartment regions (*Chen et al., 2018*). Larger amplitude, 'Type I' SON TSA-seq peaks were largely centered within A1 Hi-C subcompartments, and flanked by A2 Hi-C subcompartments, while smaller amplitude, 'Type II' SON TSA-seq peaks were contained within A2 Hi-C subcompartments (*Chen et al., 2018*).

Given the predominant correlation of differences in gene expression with differences in relative position specifically to speckles among the three nuclear locales, as described in the previous section, here we further characterize these singular iLAD/A compartment subregions. As before (*Chen et al., 2018*), operationally we identified Type I SON TSA-seq local maxima as those embedded within A1 Hi-C subcompartments versus Type II local maxima as those embedded within A2 Hi-C subcompartments (red versus blue tick marks, *Figure 3A*).

Type I peaks showed a near unimodal distribution of distance to the nearest speckle with a peak at ~100 nm distance (*Figure 3B*), based on IMR90 fibroblast multiplexed FISH data (*Su et al., 2020*). In this case we used HFF fibroblast SON TSA-seq measurements as a proxy for IMR90 fibroblast SON TSA-seq (see Section 'Polarity of Nuclear Genome Organization' for comparison of IMR90 versus HFF SON TSA-seq). In contrast, Type-II peaks showed a bimodal speckle distance distribution (Hartigan's dip test *Hartigan and Hartigan, 1985*), p-value <0.05, (*Figure 3—figure supplement 1A*), with one narrow peak similarly located at ~100 nm distance and a broader peak centered at ~500 nm from speckles (*Figure 3B*). Notably, Type II peaks show a lower speckle association frequency (<250 nm threshold, *Figure 3B*, dashed line) than Type I peaks (*Figure 3C*), suggesting that while Type II peaks still associate specifically with speckles their interaction with speckles is weaker and/or less frequent as compared to Type 1 peaks.

Both Type I (red) and Type II (blue) average peak widths are ~400 kb (*Figure 3D*); however, Type I peaks have higher speckle TSA-seq and Hi-C compartment scores (*Figure 3D*) and contain shorter genes, genes with smaller exon and intron length, and genes with lower fractional intron composition (*Figure 3—figure supplement 1B*). Across the four cell types, Type I peaks also are more conserved than Type II peak (Jaccard indexes of 0.85 vs 0.67 for Type I versus Type II; *Figure 3—figure supplement 1C*).

Although gene expression levels are similar over Type I and II peaks (*Figure 3D*, 2nd from right), the IMR90 multiplexed image data reveals a trend of gene loci within Type I peaks showing a higher fraction of 'ON' states among the cell population as compared with gene loci within Type II peaks (*Figure 3—figure supplement 1D*).

Both Type I and Type II peaks align with DNA replication initiation zones (IZs), identified as peaks in 16-fraction Repli-seq (*Figure 3A*). Type I peaks replicate slightly earlier than Type II peaks (*Figure 3E–F*, *Figure 3—figure supplement 1E*), although this difference is noticeably larger in HCT116 versus H1 cells (*Figure 3A, E and F*). Speckle TSA-seq scores progressively decrease with later replication timing of IZs (*Figure 3F*). However, DNA replication timing variability (Twidth) is notably larger for Type II peaks, particularly in HCT116 (*Figure 3E*, *Figure 3—figure supplement 1E*) suggesting a possible link between variability in speckle association (*Figure 3B*) and variability in DNA replication timing (*Figure 3E*).

We next used live-cell imaging to show that chromosome regions close to nuclear speckles show the earliest DNA replication timing; this is consistent with the earliest firing DNA replication IZs, as determined by Repli-seq, aligning with Type I peaks that are closely associated with nuclear speckles. Specifically, in HCT116 cells expressing EGFP-SON and mCherry-PCNA, the first PCNA foci appearing within the first 30 min of S-phase were nearly 100% localized within 500 nm of speckles (*Figure 3—figure supplement 1F*). From 30–135 min after initiation of S-phase, more PCNA foci appeared at increasing distances from speckles (*Figure 3—figure supplement 1F*).

In summary, these results show that iLADs are punctuated with two types of gene expression 'hot-zones' of ~400 kb width that also act as DNA replication initiation zones. Although average levels of elevated gene expression are similar among these two types of gene expression hot-zones, they contain different types of genes, different speckle association rates, and different DNA initiation zone replication timing.

## LAD to iLAD conversions mostly correspond to transitions from a repressed/ late-replicating to an intermediate repressed/ late-to-middle DNA replicating chromatin state

One important class of nuclear genome transitions are chromosome regions which undergo transitions from LADs to iLADs, as determined by their LMNB1 DamID signals. LADs and iLADs show a close correspondence to B and A Hi-C compartments, respectively, with LADs and B compartments generally associated with repressive, late replicating chromatin states versus iLADs and A

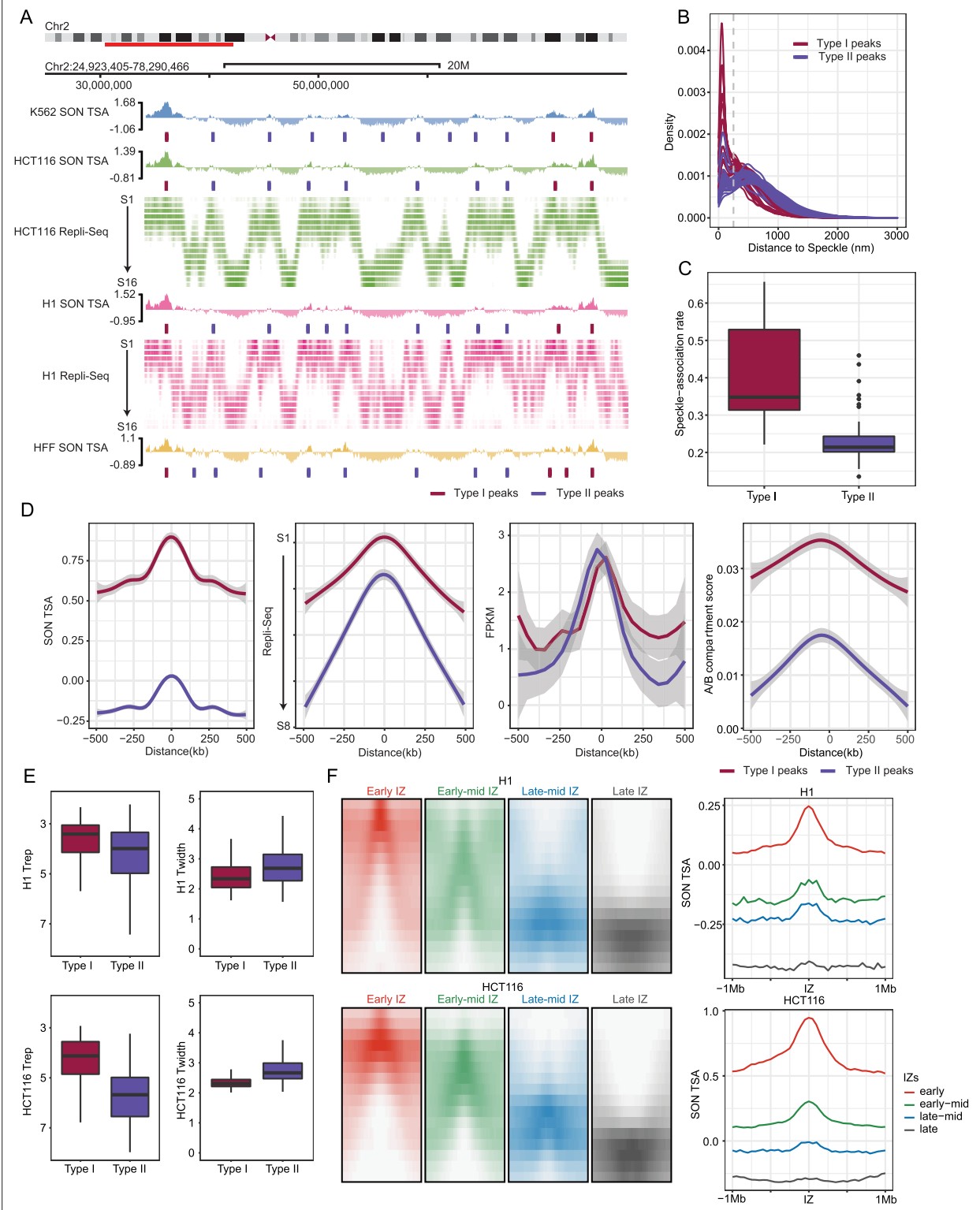

**Figure 3.** Varying gene composition, DNA replication timing, and speckle proximity of Type I versus Type II SON TSA-seq peaks which align with gene expression 'hot-zones' and DNA replication initiation zones (IZs). (**A**) SON TSA-seq Type I (red ticks) and Type II (blue ticks) local maxima ("peaks") align with DNA early replication IZs identified in 16-fraction (S1 (early) – S16 (late)) Repli-seq. Top to bottom: K562 SON TSA-seq, HCT116 SON TSA-seq and Repli-seq, H1 SON TSA-seq and Repli-seq, HFF SON TSA-seq; (**B–C**) Histograms of distances (x-axis, nm) (**B**) and boxplots showing speckle association fractions (<250 nm) (**C**) of HFF Type I and II peaks located in IMR90 fibroblasts (***Su et al., 2020***) from nuclear speckles show higher and unimodal (Type I) versus lower and bimodal (Type II) nuclear speckle attachment frequencies; (**D**) Pileup plots showing SON TSA-seq (left), Repli-seq (2nd to left), FPKM

*Figure 3 continued on next page*

*Figure 3 continued*

RNA-seq (2nd from right), and Hi-C compartment scores (right) flanking (+/-500 kb) Type I (red) versus Type II (blue) SON TSA-seq peaks; (**E**) Boxplots of Trep (timing of replication, left) and Twidth (variation in replication timing, right) in H1 (top) and HCT116 (bottom) show earlier and less variable DNA replication timing for Type I versus II peaks; (**F**) Pileup Repli-seq profiles for early, early-mid, late-mid, and late IZs (left,+/- 1 Mbp) show progressively later timing of replication correlating with the lower amplitude of SON TSA-seq peaks (right) centered at the IZ center (right).

The online version of this article includes the following figure supplement(s) for figure 3:

**Figure supplement 1.** Varying gene composition, DNA replication timing, and speckle proximity of Type I and II SON TSA-seq peaks which align both with gene expression 'hot-zones' and DNA replication initiation zones (IZs).

---

compartments which have generally been associated with active, early replicating chromatin states (*Lieberman-Aiden et al., 2009*; *van Steensel and Belmont, 2017*; *Dekker et al., 2024*). For this reason, previously it has been generally assumed that LAD to iLAD transitions signify transitions from a repressed to active chromatin state. However, the low correlation in comparisons between cell types between differences in gene expression with differences in lamina and/or nucleolar TSA-seq and lamina DamID scores (*Figure 2*, *Figure 2—figure supplement 1*) suggested that transitions from facultative LAD (fLAD) to facultative iLAD (fiLAD) might not always represent a binary transition from repressed to active chromatin.

Several previous studies have used varying approaches to subdivide LADs further into distinct subsets of LADs with different biochemical and/or functional properties (*Martin et al., 2024*; *Meuleman et al., 2013*; *Shah et al., 2023*; *Zheng et al., 2015*). However, in this Section, we focused instead on asking whether regions specifically within iLADs might show differential localization relative to the lamina and/or nucleoli and, if so, whether these regions would show different levels of gene expression. More specifically, analogously to how gene expression hot-zones appeared as local maxima in speckle TSA-seq with early DNA replication timing, we asked whether iLAD regions that appeared as local maxima in lamina proximity mapping signals would correspond to iLAD regions with locally reduced gene expression levels and later DNA replication timing relative to their flanking iLAD sequences. Our rationale was that these iLAD regions might represent chromatin domains that together with their flanking iLAD regions would typically localize well within the nuclear interior but in a fraction of the cell population would loop back and attach at the NP.

Indeed, analysis across cell lines reveals that lamina DamID scores of fLAD genomic regions can continuously transition from positive value peaks in lamina DamID (red rectangles), to peak-within-valley (p-w-v) local maxima with negative lamina DamID values (yellow rectangles), or to valleys with negative lamina DamID values (v) (blue rectangles; *Figure 4A*, *Figure 4—figure supplement 1A*). A subset of p-w-v fiLADs also showed p-w-v local maxima in their lamina TSA-seq signals together with p-w-v local maxima or small positive peaks in their nucleolus TSA-seq signals (*Figure 4—figure supplement 1A*).

We implemented an algorithm to define these three domain classes genome-wide based on their local relative increase ('enrichment') in DamID score (*Figure 4—figure supplement 1B*; see Materials and methods), revealing that p-w-v fiLADs are more common than v fiLADs in all cell lines except HFF, which has few fiLADs of either type (*Figure 4B–C*). The transitions between LADs, p-w-v fiLADs and v fiLADs are highly dynamic across the four cell lines (*Figure 4A*, *Figure 4—figure supplement 1C*), with some regions transitioning between all three domain types among the four cell lines (*Figure 4A*, *Figure 4—figure supplement 1A,C*). We observe an increasing number (*Figure 4B*) and total genome coverage (*Figure 4C*) of LADs versus p-w-v fiLADs and v fiLADs comparing H1, K562, HCT116, and HFF cell lines.

p-w-v fiLADs show late to middle DNA replication timing and reduced gene expression levels, rather than the early DNA replication timing and higher gene expression typical of constitutive iLADs (ciLADs) and a subset of v fiLADs (*Figure 4A–B*, *Figure 4—figure supplement 1D*). Moreover, transitions between cell lines from p-w-v- to v fiLADs show an increase in gene expression of comparable magnitude to that observed in LAD to p-w-v fiLAD transitions (*Figure 4*, *Figure 4—figure supplement 1*). While p-w-v fiLADs are classified as A compartment, with positive Hi-C compartment scores, in pile-up plots they show lower compartment scores than iLAD or v fiLAD regions (*Figure 4E*). Both p-w-v fiLADs and v fiLADs have earlier replication timing than LADs, but p-w-v fiLADs replicate later than v fiLADs (*Figure 4E–F*). Additionally, p-w-v fiLADs typically have an intermediate chromatin

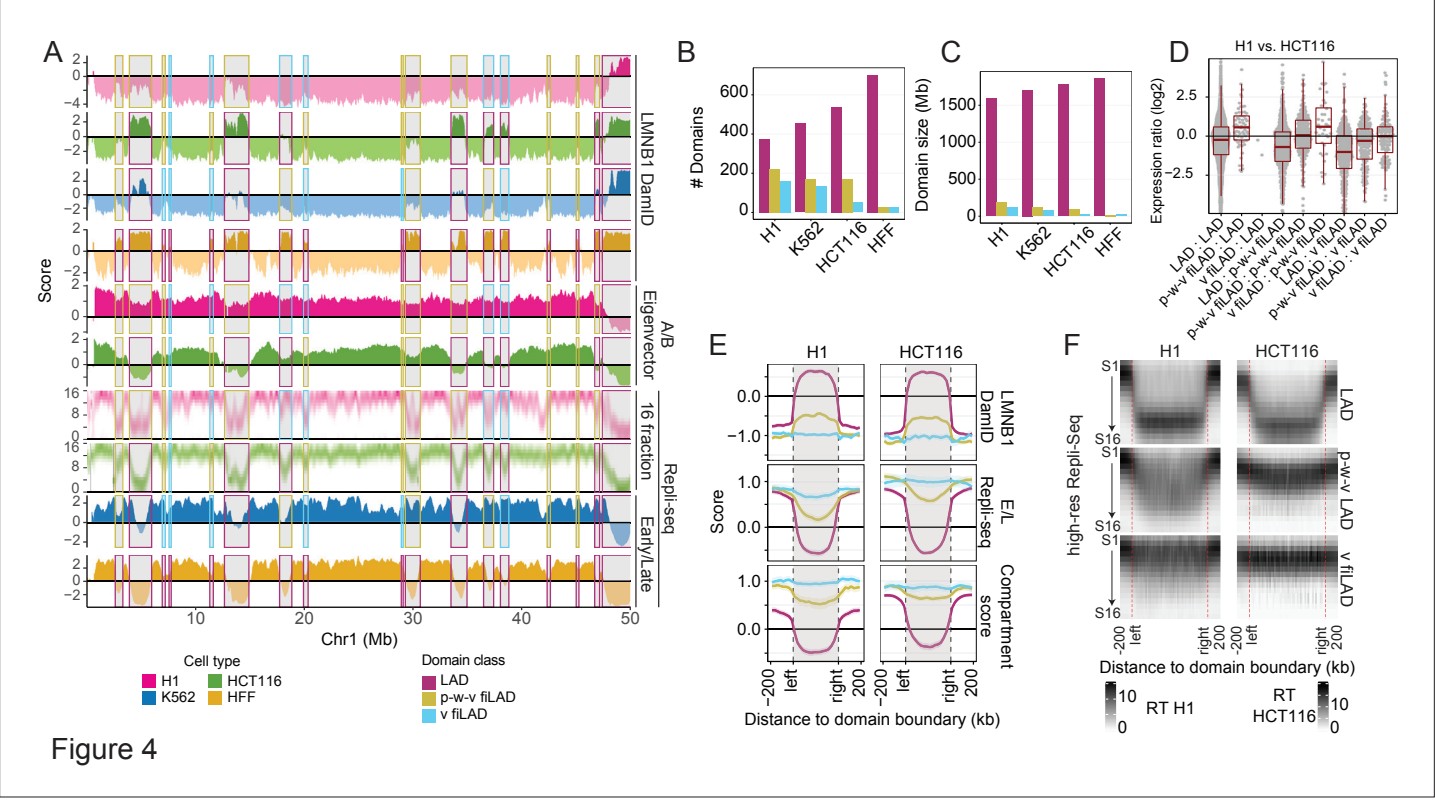

**Figure 4.** Facultative LADs transition most frequently to a partially repressed, middle-to-late replicating iLAD and less frequently to an active, early replicating facultative iLAD in different cell types. (**A**) Chr1 region with examples of LADs (red rectangles) showing peaks in LMNB1 DamID in one cell type changing in other cell types to facultative iLADs (fiLADs) showing either valleys ('v') (blue rectangles) or peak-within-valleys ('p-w-v') (yellow-orange rectangles) in their LMNB1 DamID signals. Replication timing is late for fLADs, changes to early for v fiLADs but remains late-to-middle for p-w-v fiLADs, despite Hi-C A compartment scores for both p-w-v and v fiLADs. Top to bottom - LMNB1 DamID for H1, HCT116, K562, and HFF, Hi-C A/B Eigenvector score for H1 and HCT116, 16-fraction Repli-seq for H1 and HCT116, and 2-fraction Repli-Seq for K562 and HFF (H1-red; HCT116-green; K562-blue; HFF-brown; some v fiLADs and iLADs are p-w-v fiLADs classified as v fiLADs or missed by our classification scheme); (**B–C**) Numbers (**B**) and mean sizes (**C**) of LADs (red), p-w-v fiLADs (yellow/orange), and v fiLADs (blue) in the four cell types. HFF cells have smallest number of p-w-v and v fiLADs; (**D**) Genes in same domain type show similar expression mean levels but show increased expression in p-w-v fiLADs versus LADs or v fiLADs versus p-w-v fiLADs in H1 versus HCT116. Log2 (FPKM(H1)+1 / FPKM(HCT116)+1) for genes in one type of domain in H1 and the same or another type of domain in HCT116 (H1 domain type: HCT116 domain type); (**E**) Pileup plots for LMNB1 DamID (top), E/L 2-fraction Repli-seq (middle), and Hi-C compartment score (bottom) for LADs (red), p-w-v fiLADs (yellow/orange), v fiLADs (blue) in H1 (left) versus HCT116 (right) reveals overall trend toward higher LMNB1 DamID signal, less early replication timing, and lower Hi-C compartment score of p-w-v fiLADs versus v fiLADs which resemble flanking iLAD regions; (**F**) 16-fraction Repli-seq pileup plots of LADs (top), p-w-v fiLADs (middle), and v fiLADs (bottom) confirms late, middle, versus early DNA replication patterns, respectively, for these domains in both H1 (left) and HCT116 (right).

The online version of this article includes the following figure supplement(s) for figure 4:

**Figure supplement 1.** Facultative LADs transition most frequently to a partially repressed, middle-to-late replicating iLAD and less frequently to an active, early replicating iLAD in different cell types.

state between LADs and v fiLADs, as defined by their relatively increased levels of 'repressive' and decreased levels of 'active' histone marks (*Figure 4—figure supplement 1F*).

We note that in a previous study a three-state Hidden Markov Model (HMM) segmented lamin B ChIP-seq data into two chromatin domain states with extensive overlap with LADs defined by lamina DamID (*Shah et al., 2023*). Whereas the late replicating, low gene density/expression 'T1 LAD' state showed very high overlap (98%) with LADs defined by DamID, the intermediate replicating, intermediate gene expression 'T2 LAD' state showed only 47% overlap with LADs defined by DamID. This was partly a result of the HMM segmentation algorithm but also due to substantial differences between the lamina ChIP-seq versus DamID signals for reasons that remain unclear. The subset of p-w-v iLADs included in T2 comprise only a small percentage of the total T2 LAD coverage, which includes both

other iLAD and LAD regions. Thus, the p-w-v iLADs we identified here represent a novel and distinct class of iLAD chromatin domains, not previously described.

In summary, our analysis revealed that most LAD to iLAD transitions between different cell types correspond to a transition to a partially repressive, heterochromatic state which still shows some residual association with the lamina and, in some cases, with nucleoli. These partially repressed, p-w-v ILAD regions can exist as more fully repressed LADs or more fully active iLAD chromatin domains in other cell types.

## p-w-v fiLADs compete with LADs for nuclear lamina association; linking nuclear lamina association with later DNA replication timing but not lower gene expression

In a previous section, we described how differences in gene expression showed little dependence on differences in relative distances to the nuclear lamina (*Figure 2*, *Figure 2—figure supplement 1*), challenging key predictions of previous models of genome organization. One possibility is that while many heterochromatic regions might exchange a repressive environment near the lamina for another repressive environment within the nuclear interior, a smaller number of chromosome regions might instead show a dependence of gene expression on differences in distance relative to the lamina. Indeed, we did observe that most of the ~10% of chromosome regions between cell types which showed differences in location relative to nuclear speckles also showed inverse differences in location relative to the lamina (*Figure 2—figure supplement 1D*).

Therefore, we set out to experimentally manipulate genome association with the lamina by making LMNA, LBR, and double LMNA/LBR K562 knockout (KO) lines to probe the functional consequences of changes in genome distance to the lamina. In at least some post-mitotic mouse cell types, LMNA/LBR double KO creates an 'inverted' nuclear morphology, with loss of peripheral heterochromatin, accumulation of most heterochromatin in the nuclear interior, and positioning of euchromatin toward the NP (*Solovei et al., 2013*). This inverted nuclear morphology is seen normally in retinal rod cells of nocturnal mammals, which lack both LBR and LMNA expression (*Solovei et al., 2013*), but Hi-C reveals largely unchanged A and B compartments, despite the inversion of euchromatin and heterochromatin nuclear positioning in these cells (*Falk et al., 2019*).

Whereas the LMNA KO shows little change in lamina DamID from K562 wild-type (WT) or the parental clone, the LBR KO and the LMNA/LBR double knockout (DKO) lines show similar but partial reduction in the lamina DamID for most LADs and increases for some iLADs (*Figure 5—figure supplement 1A*). Therefore, we focused our attention on comparisons between WT K562 and the DKO line.

DKO cells showed loss of the nuclear lobulation characteristic of WT K562 nuclei, the appearance of a DAPI-dense perinucleolar rim not apparent in the WT K562 nuclei, and an increased number of DAPI-dense foci in the nuclear interior (*Figure 5A*). Immunostaining against LMNB1 revealed the normal ring of staining around the NP seen in WT cells (images deposited as metadata in the deposited sequencing data sets). Based on DamID and TSA-seq, most LADs in DKO cells decrease their interaction and relative proximity with the lamina and increase their proximity to nucleoli but maintain their positioning to speckles (*Figure 5B*, *Figure 5—figure supplement 1B*); nuclear and nuclear locale morphologies change as well in the DKO cells compared to WT K562 (*Figure 5—figure supplement 1C*). Unexpectedly, most p-w-v fiLADs instead increase their lamina association while decreasing their proximity to nucleoli (*Figure 5B*, *Figure 5—figure supplement 1B, D*). Indeed, most of these p-w-v fiLADs become actual LADs in the DKO (*Figure 5—figure supplement 1E*); some v fiLADs also become LADs in the DKO (which may be due to a misclassification of v- versus p-w-v fiLADs; *Figure 5—figure supplement 1D–E*).

Scatterplots comparing changes in lamina and nucleolar TSA-seq scores between WT and DKO showed an inverse relationship (*Figure 5C*). This shift correlates with ChIP-seq data, showing that regions enriched in H3K27me3 generally shift away from nucleoli toward the lamina in the DKO, while the opposite trend is observed for regions enriched in H3K9me3 (*Figure 5D*); (consistent with the enrichment of H3K27me3 in p-w-v fiLADs versus H3K9me3 in LADs (*Figure 4—figure supplement 1F*)).

Despite the wide-spread genomic shifts relative to the lamina and nucleolus, only a small number of genes significantly change their gene expression in the KO and DKO lines (*Figure 5F*). Moreover, there is no obvious bias in the direction of gene expression changes as a function of changes in the

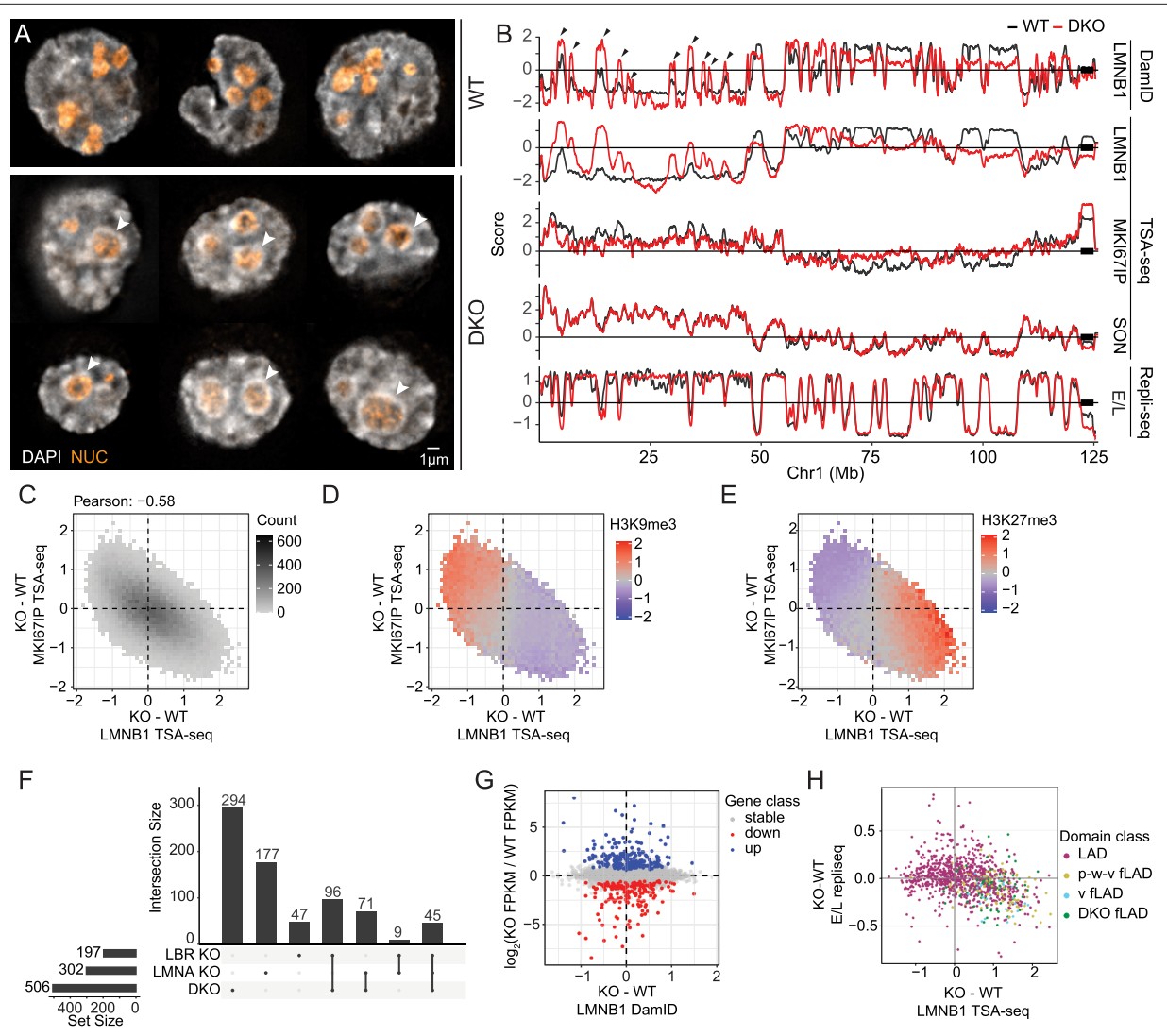

**Figure 5.** LADs shift toward nuclear interior but peak-within-valley (p-w-v) fiLADs shift toward nuclear lamina and replicate later in LMNA/LBR double knockout (DKO) K562 cells. (**A**) Representative deconvolved widefield images showing wildtype (WT) and DKO K562 cells. Scale bar = 1 μm. DNA staining (DAPI, grey) shows increased numbers of condensed chromatin foci (arrowheads) at nucleolar (orange) periphery as well as within the nuclear interior in DKO (middle and bottom rows) versus WT (top row) nuclei. DKO nuclei also are rounder and less lobular than WT nuclei; (**B**) Browser views over Chr1 left arm showing differences in (top to bottom) LMNB1 DamID, LMNB1, MKI67IP (nucleolar), and SON (nuclear speckle) TSA-seq, and 2-fraction Repli-seq (Early (E) / Late (L)) for WT (black) versus DKO (red) cells. Whereas LADs shift toward the nuclear interior, p-w-v fiLADs (arrowheads) shift toward the nuclear lamina, or become actual LADs, with slightly later replication timing; in contrast, there is no significant change in nuclear speckle positioning. (**C**) Chromosome shifts away from (toward) nuclear lamina are inversely correlated with shifts toward (away from) nucleoli: 2D- histogram showing changes (KO – WT) in LMNB1 (x-axis) versus MKI67IP (y-axis) TSA-seq. For each histogram bin (pixel), the number of overlapping 25 kb genomic bins is grey-scale coded. Only pixels with at least 10 overlapping genomic bins are plotted. Pearson correlation = –0.58; (**D–E**) H3K9me3-enriched regions shift away from nuclear periphery and toward nucleoli whereas H3K27me3-enriched regions shift toward nuclear periphery and away from nucleoli in DKO versus WT cells; 2D histograms from (**C**) overlaid with mean WT H3K9me3 (**D**) or H3K27me3 (**E**) ChIP-seq color-coded values. (**F**) Upset plot showing numbers of differentially expressed genes in each of the three KO cell lines and overlap of these differentially expressed genes between KO lines; (**G**) No trend in gene expression changes as a function of increased or decreased nuclear lamina association after DKO: Scatterplot of gene expression differences (log2-ratio FPKM, y-axis) versus differences in LMNB1 DamID (x-axis) (DKO - WT). Only active genes are shown (blue, upregulated; red, downregulated; grey, no change); (**H**) Domains that shift closer to nuclear lamina in DKO cells also shift to later DNA replication: Scatterplot comparing mean changes over chromosome domains in E/L 2-fraction Repli-seq (DKO – WT; y-axis) versus changes in LMNB1 TSA-seq (DKO – WT; x-axis). Domain data points are colored based on their transition class (see text; WT LADs, p-w-v fiLADs, and v fiLADs, and DKO fLADs).

The online version of this article includes the following figure supplement(s) for figure 5:

**Figure supplement 1.** LADs shift toward nuclear interior but peak-within-valley (p-w-v) fiLADs shift toward the nuclear lamina and replicate later after LMNA/LBR double knockout (DKO).

lamina DamID between the DKO and WT (*Figure 5G*). Similarly, there is no obvious bias toward increased gene expression among those genes contained specifically within WT LADs that change their expression and also show reduced lamina interaction (decreased lamina DamID) in the DKO (*Figure 5—figure supplement 1F*). Curiously, there is an increased fraction of differentially expressed genes in LADs versus iLADs in all KO cell lines (*Figure 5—figure supplement 1G–I*), but the significance of this bias, in the absence of bias in the direction of the changes in gene expression, remains unclear.

In contrast, plotting 2-fraction Repli-seq versus changes in DamID shows a consistent trend of slightly later DNA replication timing for regions (primarily p-w-v fiLADs) moving closer to the lamina (*Figure 5B&G*). However, LADs that shift further away from the lamina, do not show an obvious progressive shift toward earlier DNA replication timing (*Figure 5B,G,I and J*).

In summary, upon LMNA and LBR DKO in K562 cells the lamina associated heterochromatin (LADs) enriched in H3K9me3 move away NP. In contrast, heterochromatin regions localized to nuclear interior and enriched in H3K27me3 (p-w-v fiLADs) move toward the NP in DKO K562 cells. These movements show no specific dependence of gene expression on relative distances to the lamina, although a subset of chromosome regions corresponding to p-w-v fiLADs might show altered DNA replication timing as a function of increased lamina association.

## Differential positioning relative to speckles and lamina identifies different types of LAD regions

In a previous Section, we showed a large fraction of the genome showed variations in localization relative to the lamina and/or nucleoli without showing a correlation with differences in gene expression (*Figure 2*, *Figure 2—figure supplement 1*). However, these chromosome regions showed a correlation between these differences in locale localization and DNA replication timing (*Figure 2F–G*), suggesting that we might use these differences in nuclear positioning to identify different types of heterochromatic regions with differential biochemical and functional properties.

More specifically, we next asked whether we could use differences in lamina versus speckle TSA-seq scores, focusing especially on LADs, to identify heterochromatin regions with differential biochemical and functional properties (*Figure 6*, *Figure 6—figure supplement 1*).

First, whereas an overall inverse relationship between lamina and speckle TSA-seq observed in K562 cells is largely preserved in H1 cells, scatterplots reveal off-diagonal genomic bins ('H1 ROI' orange bins in *Figure 6A*, 'H1 ROI C1&C2' in *Figure 6—figure supplement 1A*) with disproportionally low lamina TSA-seq scores (*Figure 6A-B*, *Figure 6—figure supplement 1A*). These genomic bins are largely within several Mbp from centromeres (*Figure 6C*, *Figure 6—figure supplement 1E*), are H3K9me3-enriched (*Figure 6D*), and are more associated with the nucleolus specifically in H1 cells in which they are less associated with the lamina (*Figure 6—figure supplement 1B–C*) Many genomic regions contained within the H1 ROI-C1 display a similar position in the K562 DKO scatterplot (*Figure 6—figure supplement 1D*), suggesting a LMNA and/or LBR dependence for nuclear lamina association for this subgroup of LADs.

Second, deviations in the inverse relationship between SON and lamin TSA-seq in HCT116 and HFF cells with flat nuclei reveals four types of LAD regions, defined by DamID (*Figure 6—figure supplement 1F*), varying in their histone modifications (*Figure 6E–F*): (1) H3K9me3-enriched LAD Cluster C1: low SON, high LMNB1; (2) H3K27me3-enriched LAD Cluster C2: moderate SON, high LMNB1; (3) H3K9me2- and H2A.Z-enriched LAD Cluster C3: low SON, low LMNB1; (4) low in multiple histone marks LAD Cluster C4: moderate SON, low LMNB1. Superimposing these histone mark enrichments over the HCT116 scatterplot more precisely revealed the differential spatial distribution of LAD regions with different types of histone mark enrichments (*Figure 6G*).

These four types of LAD regions show different correlations with both gene expression levels and DNA replication timing (*Figure 6H*). H3K9me3-enriched C1 LAD regions show the lowest levels of gene expression. H3K27me3-enriched C2, H3K9m2/H2A.Z-enriched C3, and C4 LAD regions show gene expression levels significantly lower than iLADs but slightly higher than C1 LAD regions. DNA replication timing is latest and most uniform for the C1 LAD regions and shows progressively earlier DNA replication timing for the C3, C2, and C4 LAD regions which all still replicate later than iLADs. We defined a 'constitutive LAD (cLAD) score' as the number of cell lines out of 7 overall in which a chromosome region is a LAD. cLADs are enriched over LAD regions with the lowest SON TSA-seq

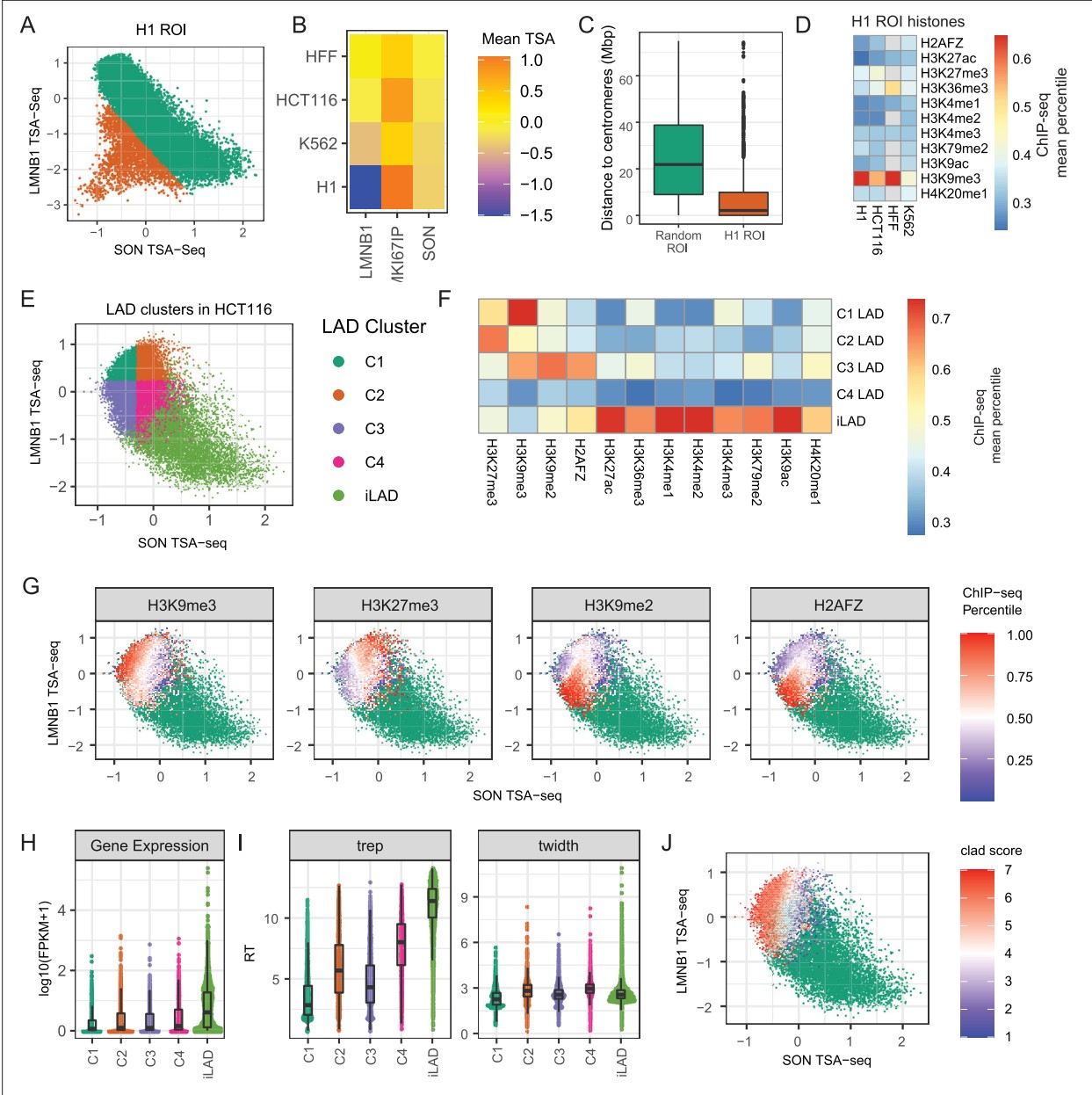

**Figure 6.** Spatial segregation within nuclei of different heterochromatin types revealed by LMNB1 and SON TSA-seq. (**A**) Genomic ROI (ROIs C1 and C2 in *Figure 6—figure supplement 1*) in H1 cells with disproportionately reduced nuclear lamina interactions (orange) which deviate from otherwise inverse correlation (green) between SON and LMNB1 TSA-seq. H1 SON (x-axis) versus LMNB1 (y-axis) TSA-seq scatterplot; (**B**) Mean TSA-seq LMNB1, MKI67IP, SON values for H1 ROI in HFF, HCT116, K562, and H1 cells (top to bottom) reveals decreased nuclear lamina but increased nucleolar proximity in H1 cells; (**C**) Most H1 ROIs are located within several Mbp of centromeres; distance (Mbp) boxplots of H1 ROIs versus other regions; (**D**) Higher enrichment of H3K9me3 versus other histone marks (mean ChIP-seq percentile values) in H1 ROIs in H1, HCT116, HFF, K562 cell lines; (**E**) Subdivision of HCT116 LAD genomic bins (100 kb) into four, color-coded clusters (C1-4) based on their SON and LMNB1 TSA-seq scores; light green points are iLAD bins; (**F**) LAD bins in C1-4 clusters show different levels of histone marks; mean percentile ChIP-seq values of indicated histone marks for the C1-4 LAD clusters and iLADs; (**G**) H3K9me3, H3K27me3, H3K9me2, and H2AFZ (H2A.Z) (left to right) enriched regions show differential nuclear localization relative to nuclear speckles and lamina roughly paralleling C1-C4 clusters; SON versus LMNB1 TSA-seq 2D histograms of LADs. The color-code represents the average histone modification levels in each bin (iLADs, green points); (**H–I**) C1-4 LAD clusters vary functionally. C1 especially but also C3 LAD bins have lower gene expression, later DNA replication timing (Trep), and more uniform replication timing (Twidth) than C2 and C4 LAD bins. iLAD bins have highest gene expression, earliest replication timing, but are less variable in replication timing than C2 and C4 LAD bins. Boxplot distributions for log2(FPKM) (**H**), Trep (**I**, left), and Twidth (**I**, right); (**J**) cLAD segregate spatially differentially from fLADs largely based on their greater distance to nuclear speckles (iLADs, green). Color-coded cLAD score (# cell lines out of seven in which a HCT116 LAD bin maps within a LAD) superimposed on SON versus LMNB1 TSA-seq scatterplot.

*Figure 6 continued on next page*

*Figure 6 continued*

The online version of this article includes the following figure supplement(s) for figure 6:

**Figure supplement 1.** LMNB1 and SON TSA-seq reveal differential intranuclear spatial segregation of centromeric and pericentromeric regions and LADs in different cell types.

scores (C1 and C3) while fLADs are enriched over LAD regions with moderate SON TSA-seq scores (C2 and C4) (*Figure 6J*).

In summary, LAD regions that segregate differentially in certain cell types relative to speckles and the lamina show different histone mark-enrichments and functional properties. Why these different types of heterochromatin segregate differentially in certain cell types remains unclear, but these results imply that such heterochromatin regions can assume very different nuclear locale positioning in different cell types.

## Polarity of nuclear genome organization

The loss of the strong inverse relationship between speckle and lamina TSA-seq in the HCT116 and HFF cell lines was unexpected and puzzling. Whereas in K562 cells the latest replicating, lowest expressing LADs enriched in H3K9me3 and cLADs map to the highest LMNB1 and lowest SON TSA-seq scores (*Chen et al., 2018*), those same cLAD regions in HCT116 cells show lower lamin B1 TSA-seq scores than fLADs enriched in H3K27me3 (*Figure 6*). What is the actual explanation for these fLADs showing even higher lamin B TSA-seq scores than cLADs in HCT116 cells but not in K562 cells? The noticeably weaker inverse relationship between speckle and lamina TSA-seq in cells with flat (HCT116, HFF) versus round nuclei (K562, H1) was due specifically to changes in lamina TSA-seq as lamina DamID scores showed less variation across cell lines (*Figure 2—figure supplement 1C*).

The inactive X chromosome, a classic example of facultative heterochromatin, was previously inferred to be localizing preferentially to the NP contained within the equatorial plane of fibroblast nuclei (*Belmont et al., 1986*). We hypothesized that LADs with the highest lamin B1 TSA-seq signal in flat HFF and HCT116 nuclei, enriched in the H3K27me3 mark (*Figure 6G*), might also correspond to genomic regions enriched specifically at the nuclear lamina contained within the nuclear equatorial plane. We hypothesized that these LADs would have the highest lamina TSA-seq scores: after lamina TSA-staining they would be exposed to a higher concentration of tryamide free-radicals generated and diffusing from the nearby side, top, and bottom of the nuclear lamina as compared to LADs at either the top or bottom of the nuclear lamina.

To test this we turned to a highly multiplexed, immuno-FISH dataset from IMR90 human fibroblasts (*Su et al., 2020*), published after our TSA-seq profiling of HFF human fibroblasts. We reasoned that the nuclear genome organization in the two human fibroblast cell lines would be sufficiently similar to justify using the IMR90 FISH dataset as a proxy for our analysis of HFF TSA-seq data. Indeed, the high inverse correlation ($R=-0.86$) of distances to speckles measured by MERFISH in IMR90 cells with HFF SON TSA-seq scores is nearly identical to the inverse correlation ($R=-0.89$) measured instead using IMR90 SON TSA-seq scores, published elsewhere (*Alexander et al., 2021*; *Figure 7A*). Similarly, distances to the nuclear lamina and nucleoli show high inverse correlations with lamina and nucleolar TSA-seq, respectively (*Figure 7A–B*). These correlations were cell type specific, particularly for the lamina and nucleolar distance correlations, as these correlations were reduced if we used TSA-seq data from other cell types (*Figure 7—figure supplement 1*). Therefore, the high correlation between IMR90 microscopic distances and HFF TSA-seq scores can be considered a lower bound on the likely true correlation, justifying the use of IMR90 as a proxy for HFF for testing our predictions.

We first simulated this effect of LADs at the equatorial plane being exposed to a higher tyramide free radical concentration during TSA staining. Convolving the HFF anti-RL1 immunostaining image with a kernel corresponding to the exponential diffusion gradient of tyramide-biotin TSA labeling (*Chen et al., 2018*), did indeed produce a notably elevated TSA signal in the equatorial plane of the NP (*Figure 7C*). These simulation results predict that the LADs with the highest lamina TSA-seq scores lie within the equatorial plane, while LADs that show lower lamina TSA-seq scores, but similar lamina DamID scores, either distribute over the NP randomly or are biased toward the top or bottom of the nucleus.

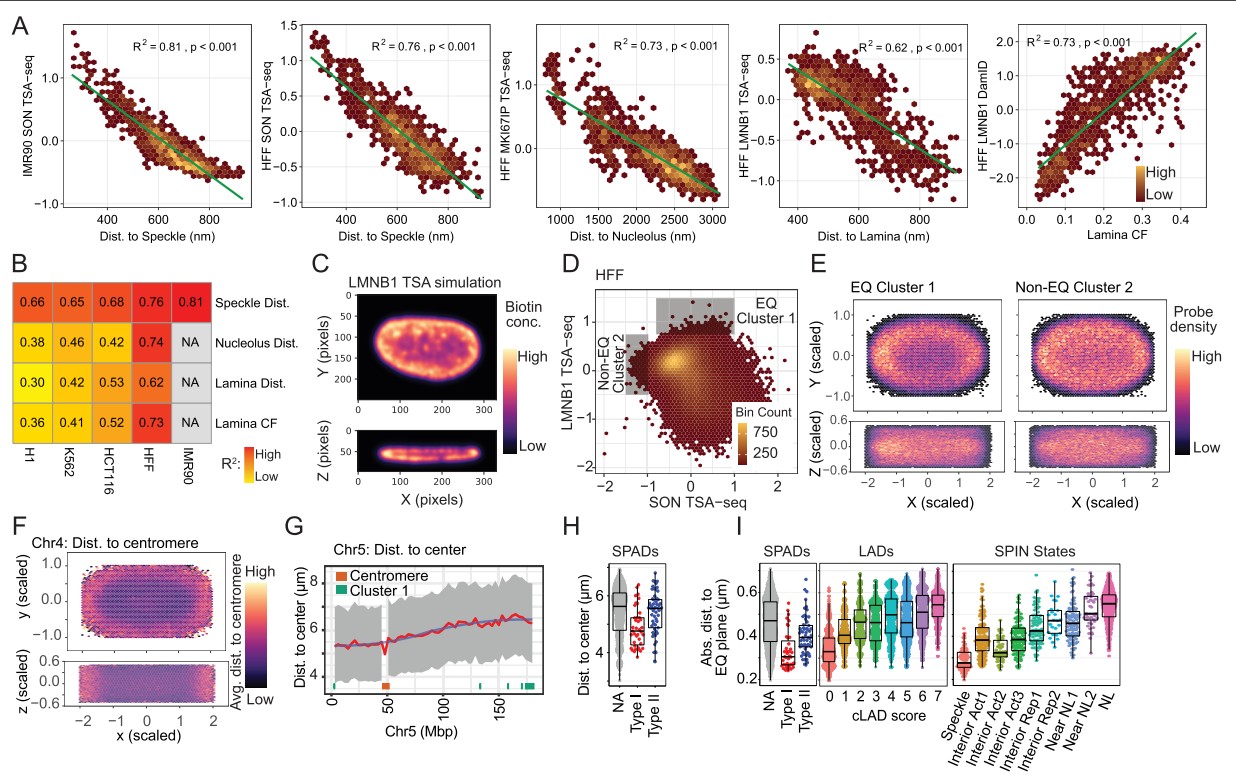

**Figure 7.** Polarity of nuclear genome organization. (**A–B**) Microscopic distances measured by MERFISH (*Su et al., 2020*) inversely correlate with TSA-seq scores while lamina contact frequencies measured by MERFISH correlate with lamina DamID. (**A**) Left to second to right: Scatterplots showing distances of FISH probes to nuclear locales (x-axis) in IMR90 fibroblasts versus TSA-seq scores (y-axis) for speckles in IMR90, for speckles, nucleoli (MKI67IP), and lamina (LMNB1) in HFF fibroblasts; right panel: lamina contact frequencies measured by microscopy in IMR90 (x-axis) versus lamina (LMNB1) DamID score (y-axis); (**B**) Best correlations are seen for comparisons of IMR90 data with the similar fibroblast line HFF as compared to other cell lines: correlation coefficients (R²) for comparison of MERFISH distances and contact frequencies (lamina) with TSA-seq or DamID measured in different cell lines; (**C–E**) LMNB1 TSA-seq provides a readout of nuclear genome polarity in flat nuclei due to the diffusion radius of tyramide free-radicals, identifying a LAD subset preferentially localizing at the nuclear equatorial plane periphery. (**C**) Nuclear pore immunostaining from HFF nucleus convolved with the 3D exponential decay function of TSA staining (pseudo-colored intensity) predicts higher biotin-labeling of lamina-associated chromatin lying in nuclear equatorial plane versus top or bottom of nucleus. Top (x-y cross-section); bottom (x-z cross-section). Pixel size = 80 nm; (**D**) 2D color-coded histogram showing number of LAD genomic bins with given SON (x-axis) and LMNB1 (y-axis) TSA-seq mean values in HFF. Rectangular boxes show selection of EQ Cluster 1, with the highest LMNB1 TSA-seq values, and the Non-EQ Cluster 2, with low-to-moderate LMNB1 and low SON TSA-seq; (**E**) Cluster 1 LADs preferentially localize to the equatorial plane of the nuclear periphery. Heat map showing FISH probe location density over many IMR90 fibroblast nuclei from Cluster 1 (left) versus Cluster 2 (right) LADs superimposed on normalized nuclear shape. Top: x-y projection; bottom: x-z projection; (**F**) Distal chromosome arms also preferentially localize to equatorial plane. Same as (**E**) but using all the FISH probes mapped to Chr4 and color-coded by distance (averaged over probes) to centromere in Mbp; (**G**) Distance to IMR90 nucleus x-y center also varies with chromosome distance from centromeres: IMR90 FISH probe distances (y-axis) from nuclear center (projected x-y plane distances) as a function of Chr5 position (x-axis). Grey- interquartile range of probe distances; blue- mean probe distance; red- smoothed mean probe distance. Bottom track- red rectangle marks centromere position, green rectangles mark Cluster 1 LADs.; (**H and I**) Overall nuclear genome polarization in fibroblasts (IMR90); (**H**) Boxplots of mean distance to nuclear center (projected x-y plane distance) for all probes (NA), or Type I (red) or Type II (blue) SON TSA-seq peaks; (**I**) Boxplots of mean distance to equatorial plane for SON TSA-seq peaks, facultative versus constitutive LADs, and SPIN states. Strong versus moderate bias of Type I (red) versus Type II (blue) HFF SON TSA-seq peaks, respectively, to locate close to x-y nuclear center (**H**) and near to the equatorial plane of nuclear interior (I, left panel). Facultative LADs (low cLAD scores) localize closer to equatorial plane than constitutive LADs (high cLAD scores) (I, middle panel). SPIN states show progressive trend of increased mean distances to equatorial plane from 'speckle' to 'interior active' to 'interior repressed' and 'near lamina 1', to 'near lamina 2' and 'lamina' SPIN states (*Wang, 2021*) (I, right panel).

The online version of this article includes the following figure supplement(s) for figure 7:

**Figure supplement 1.** IMR90 fibroblast MERFISH data serves as a good proxy for HFF fibroblast genomic data.

**Figure supplement 2.** Polarity of nuclear genome organization.

To test these predictions, we identified LADs with very different lamina TSA-seq scores, measured in human HFF fibroblasts, and compared their microscopic localization within IMR90 human fibroblast nuclei, measured by immuno-FISH.

Specifically, we compared the localization in HFF cells of 26 LADs mapping to regions of highest lamina TSA-seq (EQ Cluster 1, *Figure 7D*) versus 27 LADs mapping to regions of lowest speckle TSA-seq but with low-to-moderate lamina TSA-seq (Non-EQ Cluster 2, *Figure 7D*). Cluster 1 LADs showed a biased localization in IMR90 fibroblasts toward the equatorial plane as compared to Cluster 2 LADs that distributed more uniformly over the x-y nuclear projection (*Figure 7E*). More generally, Cluster 1 and 2 LADs show a similar differential localization in lamina versus SON TSA-seq scatterplots in both HFF and HCT116 cells with flat nuclei which is notably different than in K562 and H1 cells with round nuclei (*Figure 7—figure supplement 2A*). Cluster 1 LADs correspond to LADs located toward the ends of long chromosome arms (*Figure 7—figure supplement 2B*). Overall, in fibroblasts there is a trend of distal chromosome arms localizing preferentially toward the nuclear equator versus centromeric regions localizing preferentially toward the nuclear center (*Figure 7F–G* and *Figure 7—figure supplement 2C–D*).

We next examined the nuclear polarity of genomic regions associated with different nuclear locales in fibroblasts. Single-cell data (*Figure 7—figure supplement 2E*) and average trends over all nuclei (*Figure 7—figure supplement 2F*) show speckle-associated regions concentrated toward the equatorial plane, consistent with the direct localization of speckles within the equatorial plane (*Figure 1B*). Nucleolus-associated regions were distributed toward the nuclear center but throughout the z-plane (*Figure 7—figure supplement 2E–F*). Type I speckle attachment regions localize more centrally in the x-y plane and closer to the equatorial plane than Type II speckle attachment regions (*Figure 7H–I*, *Figure 7—figure supplement 2G*). The more peripheral, less central positioning of Type II peaks is consistent with the shorter genomic distances of Type II versus I peaks to the nearest LADs (*Chen et al., 2018*). fLADs localize closer to the equatorial plane (*Figure 7I*) consistent with the enrichment of H3K27me3 modification in these regions, while cLADs localize furthest away from this plane (*Figure 7I*). Additionally, v fiLADs cluster closer to the equatorial plane and further from the nuclear center as compared to p-w-v fiLADs (*Figure 7—figure supplement 2H*).

To more comprehensively survey how the genome distributes relative to nuclear polarity, we turned to the use of SPIN states, which combine TSA-seq and DamID with Hi-C data to segment the genome into 9–10 states based on their intranuclear localization; SPIN states show differential epigenetic marks, gene expression, and DNA replication timing (*Wang, 2021*; *Wang et al., 2021*). SPIN states ordered according to highest to lowest gene expression, and earliest to latest DNA replication timing, showed a pronounced, nearly monotonic increasing distance from the equatorial plane (*Figure 7I*) as well as differential localization relative to the x-y nuclear center (*Figure 7—figure supplement 2I*).

Thus, in adherent cells with flat nuclei the distance from the equatorial plane emerges as an additional key axis in nuclear genome organization in addition to the x-y plane radial position.

## Discussion

Here, we integrated imaging with both spatial (DamID, TSA-seq) and functional (Repli-seq, RNA-seq) genomic readouts across four human cell lines. Overall, our results significantly extend previous nuclear genome organization models, while also demonstrating a cell-type-dependent complexity of nuclear genome organization. Briefly, in contrast to the previous radial model of genome organization, we reveal a primary correlation of gene expression with relative distances to nuclear speckles rather than radial position. Additionally, beyond a correlation of nuclear genome organization with radial position, in cells with flat nuclei we show a pronounced correlation of nuclear genome organization with distance from the equatorial plane. In contrast to previous binary models of genome organization, we describe how both iLAD / A compartment and LAD / B compartment contain within them smaller chromosome regions with distinct biochemical and/or functional properties that segregate differentially with respect to relative distances to nuclear locales and geometry.

More specifically, we describe five significant new insights into genome organization (*Figure 8*). First, across the four cell lines examined, differences in gene expression correlate primarily with differences in the genome positioning of these genes relative to nuclear speckles (*Figure 8A*) rather than distance from the nuclear lamina and despite variations in speckle positioning relative to the nuclear lamina and nucleoli. Consistent with previous observations showing a more interior nuclear localization

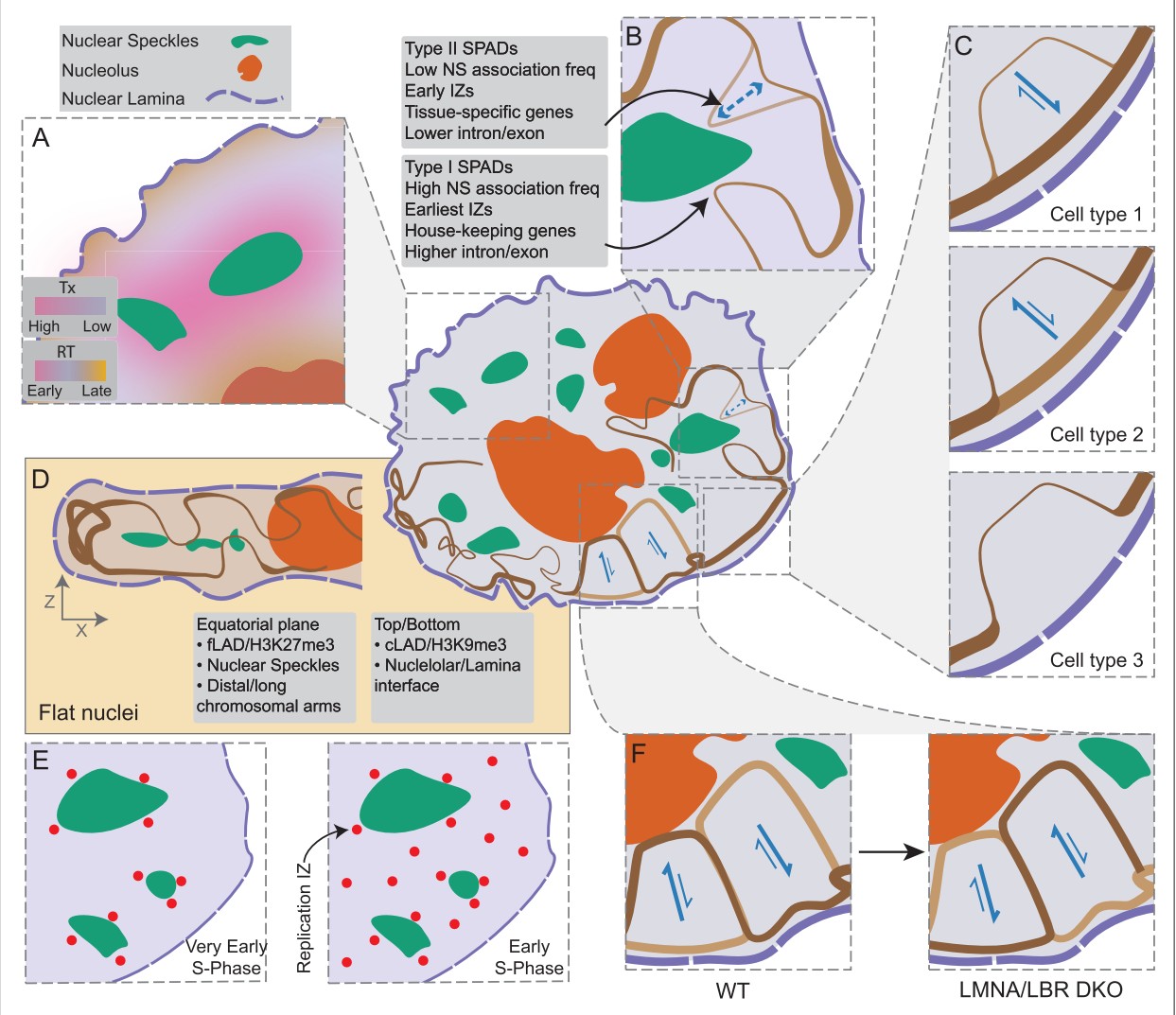

**Figure 8.** New insights into nuclear genome organization revealed by nuclear locale genome mapping. Human nucleus schematic (middle) shows nuclear speckles (NS, green), nucleolus (orange), and nuclear lamina (NL, purple). (**A**) Changes in the transcriptional activity of genes (pink to purple gradient) primarily correlates with changes in their distance to nuclear speckles (green), rather than nuclear lamina or nucleolus, whereas changes in DNA replication timing (pink-purple-yellow gradient) correlates with distance to all three nuclear locales. Regions close to nuclear speckles replicate early, while regions close to nucleolus and/or nuclear lamina replicate late during S-phase. (**B**) Two types of speckle association domains (SPADs), with similarly elevated levels of gene expression but genes of different lengths and relative intron/exon fraction, associate with NS. Type-I SPADs, with shorter, more exon-rich genes, have higher NS association frequencies compared to Type-II SPADs, with longer, more intron-rich genes. (**C**) fLADs can assume at least three different chromatin states across different cell types: (i) a B-compartment LAD (positive lamina DamID signal) with high NL association, low gene expression levels and late DNA replication (cell type 1); (ii) an A-compartment p-w-v fiLAD (peak-within-valley lamina DamID signal) with low NL association, intermediate gene expression and middle-to late DNA replication (cell type 2); (iii) an A-compartment v fiLAD (valley lamina DamID signal) with no NL association, high gene expression and early DNA replication (cell type 3). (**D**) In cells with flat nuclei, the genome is differentially organized relative to the nuclear equatorial plane: facultative heterochromatin regions are enriched near the NL within the equatorial plane, while constitutive heterochromatin regions are enriched near the NL toward the top or bottom of the nucleus. Nuclear speckles localize toward the nuclear equatorial plane and toward the nuclear center, with Type I SPADs closer than Type II SPADs to the equatorial plane and nuclear center. (**E**) Type I and II SPADs align with DNA replication initiation zones with the earliest-firing DNA replication initiation zones mapping to Type I SPADs and the earliest DNA replication foci appearing adjacent to NS. (**F**) LADs and p-w-v fiLADs compete for NL versus nucleoli association: in a LMNA/LBR double knockout cell line, LADs shift from the NL toward nucleoli and the nuclear interior while p-w-v fiLADs shift toward the NL.

of genes after their activation (**Bickmore, 2013**; **Takizawa et al., 2008**), we demonstrate an inverse trend between changes in speckle and lamina distances for the small fraction of genomic regions that show significant changes in their relative speckle distances. For these regions, there is a strong bias toward increased (decreased) gene expression with decreased (increased) speckle distance. However,

several-fold more genomic regions significantly change their relative lamina distances without significantly changing their speckle distances or gene expression. Similarly, LBR/LMNA DKO in K562 cells results in many genomic regions increasing or decreasing their lamina association without changes in their speckle distance or gene expression. In contrast, changes in DNA replication timing correlate with changes in positioning relative to all three nuclear locales – speckles, lamina, and nucleoli (*Figure 8A*).

Second, Hi-C A compartment/iLADs are punctuated by strong (Type I) and weaker (Type II) speckle attachment regions which align with DNA replication initiation zones and show elevated gene expression levels above flanking iLAD regions (*Figure 8B*). Type I regions contain shorter, more exon-rich genes as compared to Type II regions and are more highly conserved across cell lines than Type II regions. IZs within strong speckle attachment sites replicate earlier than IZs within weak speckle attachment zones with the very earliest DNA replication foci appearing adjacent to nuclear speckles (*Figure 8E*).

Third, Hi-C A compartment/iLADs are also punctuated by partially repressed fiLADs which retain weak lamina interaction and, in some cases, elevated nucleolar interactions. Identified as local maxima in lamina TSA-seq and/or DamID, these "p-w-v fiLADs" comprise ~2/3 of all fiLADs across the four cell lines examined. P-w-v fiLADs show gene expression levels intermediate between LADs and typical iLADs while retaining late or shifting to middle DNA replication timing. Only approximately 1/3 of fiLADs, identified as valleys in lamina DamID (v fiLADs), convert to an active, early-replicating, interior chromatin state. The same fLADs across different cell lines can associate strongly with the lamina as a LAD, retain weak lamina association as a partially repressive p-w-v fiLAD, or show a true transition to an interior, active, early DNA replicating state (*Figure 8C*). Upon LMNA/LBR double KO, p-w-v fiLADs shift closer to the lamina, suggesting they compete with LADs for lamina association in WT cells (*Figure 8F*).

Fourth, we show how LAD regions showing different histone marks- either enriched in H3K9me3, H3K9me2 plus H2A.Z, H3K27me3, or none of these marks- can differentially segregate within nuclei. These results support the previous suggestion of different 'flavors' of LAD regions, based on the sensitivity of the autonomous targeting of BAC transgenes to the lamina to different histone methyltransferases (*Bian et al., 2013*). Differential nuclear localization also was recently inferred by the appearance of different Hi-C B-subcompartments, which similarly were differentially enriched in either H3K9m3, H3K27me3, or the combination of H3K9me2 and H2A.Z (*Spracklin et al., 2023*). More recently, and while this paper was in revision, a new study described segmenting mouse embryonic fibroblast LADs into three clusters using histone modification profiling (*Martin et al., 2024*). Interestingly, these three LAD clusters also most notably differed by their dominant enrichment of either H3K9me3, H3K9me2, or H3K27me3. Thus, three orthogonal approaches have converged on identifying different LAD regions showing differential enrichment either of H3K9me3, H3K9me2, or H3K27me3. Here, our use of TSA-seq directly measures and assigns the intranuclear localization of these different LAD regions to different nuclear locales.

Fifth, beyond the previously described dependence of genome organization relative to the radial axis in the x-y plane, there exists an unexpectedly high degree of nuclear polarity with respect to an orthogonal z-axis in HCT116 and HFF cells with flat nuclei (*Figure 8D*). Strong speckle attachment regions localize closer to the equatorial plane than weak speckle attachment regions. A subset of LADs, primarily at the ends of long chromosome arms, preferentially localize at the lamina within this equatorial plane. Using SPIN states, ordered by DNA replication timing and levels of gene expression, we show that this overall nuclear polarity applies to nearly the entire genome, suggesting a corresponding overall gradient in both later DNA replication timing and lower gene expression with increased distance from the equatorial plane.

In conclusion, by examining genome organization relative to multiple nuclear locales, we were able to demonstrate a higher correlation than previously apparent between nuclear location and functions such as DNA replication timing and gene expression. We anticipate future mapping of the genome relative to additional nuclear locales will reveal a still more deterministic relationship between nuclear positioning and genome function.

## Materials and methods

### Cell culture

H1, HCT116, and HFF cells were cultured according to standard operation protocols established by the NIH 4D Nucleome Consortium (https://www.4dnucleome.org/cell-lines.html). K562 cells were cultured according to the ENCODE Consortium protocol (http://genome.ucsc.edu/ENCODE/protocols/cell/human/K562_protocol.pdf).

### Immunostaining

Adherent cells (HCT116, HFF, and H1) were seeded on 12 mm round coverslips and cultured for two days prior to immunostaining. For K562 cells, 150 µl of cell suspension was added to coverslips treated with poly-L-lysine as described previously (*Chen et al., 2018*). All washing steps were 3×10 mins in Ca, Mg- free PBS at room-temperature (RT), unless specified elsewise. Cells were fixed with freshly made 2% paraformaldehyde (PFA) in PBS for 10 min at RT and washed. Cells were permeabilized for 5 min on ice with 1 X PBS containing 0.2% Triton X-100 (Sigma-Aldrich T8787) and 1% natural goat serum (NGS) and washed with 1XPBS/1% NGS (PBS/NGS) 3x10 min at RT. Each coverslip was then incubated with 100 µl of the primary antibody cocktail in a dark humid chamber at 4 °C overnight. The cocktail contained moue-anti-RL1 (Santa Cruz cat# sc-58815, 1:100 dilution), rabbit-MKI67IP (Atlas antibodies cat# HPA035735, 4 ng/ml dilution) and rabbit-anti-SON (Pacific Immunology Corp, custom-raised, 4 ng/ml dilution; *Chen et al., 2018*) diluted in PBS/NGS. The anti-MKI67IP and anti-SON antibodies were directly labelled with CF-594 and CF-640R fluorophores using Mix-n-Stain CF Dye Antibody Labeling Kits (Biotium cat #92256 and #92258) according to manufacturer's instructions. After immunostaining, coverslips were washed and RL1 antibody was immunostained with goat-anti-mouse-Alexa-Fluor-488 secondary antibody (Jackson ImmunoResearch Laboratories, cat#115-545-003 diluted 1:500 in PBS/NGS) for 1 hr in a dark humid chamber at RT and washed. The DNA was counterstained with 200 ng/ml DAPI in PBS for 10 mins at RT and washed. Finally, the coverslips were mounted on glass slides using an antifade mounting media (10% w/v Mowiol 4-88(EMD Millipore)/1% w/v DABCO (Sigma-Aldrich)/25% glycerol/0.1 M Tris, pH 8.5) and cured overnight before proceeding to imaging.

### Microscopy and image segmentation

Locales were imaged using the OMX-V4 microscope (GE Healthcare) equipped with a U Plan S-Apo 100×/1.40-NA oil-immersion objective (Olympus), two Evolve EMCCD cameras (Photometrics). Z-sections were 125 nm apart. Images were segmented in 3D using a combination of in-house softwares and the Allen Cell and Structure Segmenter (Segmenter) toolkit in Python (*Chen et al., 2020*). Briefly, nuclei were first identified by segmenting DAPI using Otsu thresholding method. RL1 signal was first segmented using a modified nup153 workflow from Segmenter. In H1 cells, there are some cytoplasmic signals for RL1 which were removed using the DAPI nucleus mask. RL1 signals are puncta and the alpha-shapes algorithm (*Edelsbrunner et al., 1983*) was used to calculate a 3D mesh-surface enclosing all the puncta (*Figure 1—figure supplement 1A*). This surface was used to measure the nuclei's volume (V), surface area (SA), and SA/V. We segmented SON using the SON workflow, and MKI67IP using 'masked object thresholding' workflow from Segmenter. The binary segmented images were used to measure the total number of locales in each nucleus and average and total V, SA. Nuclei images and segmentations were manually curated for quality control and are available online. To measure locale distances in each cell line, the SON and MKI67IP binary images were converted to mesh surfaces using a marching cube algorithm. An asymmetric locale to locale distance was then calculated by measuring the closest distance between a pair of vertices from locales mesh-surfaces (*Figure 1—figure supplement 1B*). The PCA analysis was performed in R using 15 numerical metrics for each nucleus (*Figure 1F*, *Figure 1—figure supplement 1*). A similar approach was used to compare K562 WT and LMNA/LBR double knockout cells.

### HCT116 EGFP-SON KI

A Cas9/sgRNA RNP complex and linear dsDNA donor were transfected into HCT116 cells using the Amaxa Nucleofector II device (Lonza) set for program D-032 with the Cell Line Nucleofector Kit V. We obtained the guide and CRISPR RNA using the GeneArt Precision gRNA Synthesis Kit (Thermo Fisher Scientific) with our designed SON guide sequence 5'- gagagaacggagcggacgcca –3'. The Cas9 protein Alt-R S.p. Cas9 Nuclease V3 was purchased from Integrated DNA Technologies (IDT). Donor DNA was

amplified by PCR from a previously modified version of the SON-containing BAC RP11-165J2 (Invitrogen) with EGFP sequence at the NH2 terminus of SON (*Khanna et al., 2014*). We used modified primers containing a 5' amine group with a C6 linker (*Yu et al., 2020*) 5'- GTGCTCACTGATTGGT CCCTC - 3' and 5' - GAACGACTGCGTCTCCGAAG - 3' (amC6, IDT) to obtain a 1044 bp fragment that includes the EGFP sequence flanked by a 160 bp upstream and a 165 bp downstream homologous region of SON. After 5 days of recovery, EGFP-positive cells were sorted using the FACS ARIA II sorter at the UIUC Roy J. Carver Biotechnology Center. Individual HCT116 clones expressing SON-GFP were genotyped by PCR with primers 5'- CAAGAGAGACGGCTCCTGTAATG –3' and 5'- TCGG AAAAGGCGAAGTTCCTCG –3' which amplify the complete EGFP integration to identify the double allele knock-in clone "C4". HCT116 EGFP-SON clone C4 was transduced with a lentivirus carrying the Lenti-mCherry-PCNA construct (*Xiong et al., 2023*) by supplementing media with 8 μg/ml polybrene (Santa Cruz Biotechnology).

## Live-cell Imaging and analysis

HCT116 EGFP-SON clone C4 cells expressing mCherry-PCNA were seeded in 35 mm dishes with a #1.5-thickness glass coverslip bottom (MatTek, P35G-1.5–14 C) and grown to ~90% confluency. Live cell imaging was performed using a OMX V4 (Applied Precision) microscope with a 100×/1.4 NA oil immersion objective (Olympus), two Evolve EMCCDs (Photometrics), and a live-cell incubation chamber. The chamber contains a temperature control (37 °C) for the incubator and the objective lens, and a humidified CO2 supply. 3D images were acquired using a z-spacing of 300 nm once every 5 mins for 12 hr. Each z-slice was imaged at 2.0% transmittance and 10ms exposure for 488 nm excitation (GFP, Speckles), and 5.0%T with 10ms exposure for 568 nm (mCherry, PCNA). We used FIJI to smooth images and remove noise. A median filter of 0.24 μm and 'rolling ball' background subtraction of ~0.5 μm was used. All measurements were done from corresponding individual Z-sections per timepoint. To identify PCNA foci we used the FIJI maximum entropy threshold plugin followed by watershed segmentation of the grayscale image. FIJI's particle analysis was used to identify the center of each outlined PCNA foci, and the distance to nuclear speckle was measured from the PCNA foci center to the visible edge of the nearest nuclear speckle.

## K562 LMNA and LBR single and double knock-out generation

To create LMNA and LBR knockout (KO) lines and the LMNA/LBR double knockout (DKO) line, we started with a parental 'WT' K562 cell line, clone #17, expressing an inducible form of Cas9 (*Brinkman et al., 2018*). The single KO and DKO were generated by CRISPR-mediated frameshift mutation according to the procedure described previously (*Schep et al., 2021*). The 'WT' K562 clone #17 was used for comparison with the KO clones. The LBR KO clone, K562 LBR-KO #19, was generated, using the LBR2 oligonucleotide GCCGATGGTGAAGTGGTAAG to produce the gRNA, and validated previously, using TIDE (*Brinkman et al., 2014*) to check for frameshifts in all alleles as described elsewhere (*Schep et al., 2021*). The LMNA/LBR DKO, K562 LBR-LMNA DKO #14, was made similarly, starting with the LBR KO line and using the combination of two oligonucleotides to produce gRNAs: LMNA-KO1: ACTGAGAGCAGTGCTCAGTG, LMNA-KO2: TCTCAGTGAGAA GCGCACGC. Additionally, the LMNA KO line, K562 LMNA-KO #14, was made the same way but starting with the 'WT' K562 cell line. Validation was as described above; additionally, for the new LMNA KO and LMNA/LBR DKO lines, immunostaining showed the absence of anti-LMNA antibody signal under confocal imaging conditions used to visualize the WT LMNA staining while the RNA-seq from these clones revealed an ~20-fold reduction in LMNA RNA reads relative to the WT K562 clone.

## TSA-seq

SON and MKI67IP TSA-Seq was performed using Condition E (labeling with 1:300 tyramide biotin, 50% sucrose and 0.0015% hydrogen peroxide) (*Zhang et al., 2021*) with the following minor modification: 150 μl of Dynabeads M-270 streptavidin (Invitrogen, catalog no. 65306) was used to purify the biotinylated DNA. For LMNB1 TSA-Seq, Condition A2 (*Zhang et al., 2021*) (labeling with 1:10000 tyramide biotin, 50% sucrose, 0.0015% hydrogen peroxide and reaction time 20 min at RT) was used.

## DamID-seq and data processing

The DamID-seq and subsequent processing was performed according to a previously published protocol (*Leemans et al., 2019*).

## RNA-seq (for K562 DKO)

RNA was isolated using the Qiagen RNeasy column purification kit, after which sequencing libraries were prepared using the Illumina TruSeq polyA stranded RNA kit.

## Repli-seq and data processing (for K562 DKO)

The 2-fraction Repli-seq used in analysis of the K562 DKO cell line was done according to a previously published protocol (*Marchal et al., 2018*). Briefly, BrdU was added to a final concentration of 100 µM and cells were incubated at 37 °C for another 2 hr. Cells were fixed in ice-cold ethanol and processed for Repli-seq as described before.

## Genomic segmentations

For detecting SON TSA-seq Type I and Type II peaks, the SON TSA-seq score (25 kb bins) first was smoothed using locally weighted regression (LOESS). Next, a local maximum filter with window size (w) was applied, which recorded the maximum SON TSA-seq score within this window for each genomic locus. Then the data after the maximum filter were subtracted from the original smoothed data. Peaks were retained only if they showed a reduction in value, which indicates true local peaks. A parameter optimization procedure was conducted in order to maximize the peak agreement between two SON TSA-seq replicates. As a result, a smoothing factor (span = 0.005) and a window size (w=50) were selected. To distinguish local peaks into Type I and Type II, we overlapped them with Hi-C subcompartments (*Rao et al., 2014*). Genomic loci overlapping A1 subcompartments were classified as Type I peaks, while those overlapping A2/B1 were designated as Type II peaks.

ROIs in H1 cells corresponding to off-diagonal 100 kb genomic bins in SON versus LMNB1 TSA-seq scatterplots were defined as those bins below the line LMNB1=−1.5× SON – 1.3, in these scatterplots. H1 ROI genomic regions were further subdivided into H1 ROI-C1 and C2 sub-regions (*Figure 6— figure supplement 1*), in which the H1 ROI-C2 regions were defined as the genomic bins below the LMNB1=−1.5× SON – 2.2 line, while ROI-C1 regions were all remaining ROI bins.

To define LADs, p-w-v fiLADs, and v fiLADs, first we identified a consensus set of all chromosome regions which are LADs in at least one of the four examined cell types. Segmentation into these three domain classes was implemented heuristically using the local relative increase ('enrichment') of the domain DamID score relative to its flanking regions (*Figure 4—figure supplement 1*): LAD domains, defined by their positive DamID scores using a Hidden Markov model, showed the largest positive local enrichment. We defined v fiLADs as those domains which changed from LADs in a different cell line to iLADs and which showed lower local enrichment than 95% of the LADs. P-w-v fiLADs were then defined as all other fiLAD domains and showed intermediate local enrichment levels.

To define LAD clusters in HCT116 cells based on LMNB1 vesus SON TSA-seq scatterplots, all 100 kb genomic bins corresponding to LADs based on the HCT116 DamID (4DN file) were divided into 4 quadrants (*Figure 6E*) centered at SON = −3.0 and LMNB1=0.25. The C1, C2, C3, and C4 clusters were defined as the top-left, top-right, bottom-left, and bottom right quadrants, respectively.

Equatorial (Cluster 1) and non-equatorial (Cluster 2) LADs were defined in HFF cells (*Figure 7B*) as those 100 kb bins which were LADs as defined by the LMNB1 DamID which had LMNB1 TSA-seq (4DN file number) scores higher than 0.75 for the equatorial LADs or SON TSA-seq (4DN file number) scores less than 0.98 for the non-equatorial LADs.

## Gene expression analysis

Regarding *Figure 6H*, the genes overlapping with C1-C4 LAD clusters were identified and their expression was evaluated in HCT116 RNA-seq.

## Correlations with other genomic features

Data sets used for this study (*Supplementary file 1*) include TSA-seq, DamID-seq, Repli-seq, RNA-seq, ChIP-seq and HiC compartment scores. TSA-seq, DamID, Repli-seq, and ChIP-seq data were

mapped to hg38 and binned in 25 kb genomic windows, except for analysis included in *Figure 6*, *Figure 6—figure supplement 1*.

For the ChIP-seq analysis in *Figure 6*, fold-change over input was calculated for 25 kb bins and was converted to percentile values across the genome. The percentile values were averaged over all the bins corresponding to the C1-C4 LAD clusters in the *Figure 6F* heatmap or were overlayed on the LMNB1 versus SON TSA-seq binned scatterplots in *Figure 6G*.

A cLAD score was calculated using the DamID LAD calls in seven available human cell lines (*Supplementary file 1*). First, the union of all LAD calls among the seven cell lines was computed. For each 100 kb genomic bin, a number between 1 and 7 was assigned depending on the number of cell lines in which the bin was called as a LAD. The average cLAD score was overlayed on the LMNB1 versus SON TSA-seq binned scatterplot in *Figure 6J*.

## Correlating omics data with the IMR90 multiplex FISH data

For the LMNB1 TSA simulation (*Figure 7A*), a 3D image of the segmented HFF NP (anti-RL1 staining) was used. The binary image was convolved with a symmetrical 3D kernel based on the exponential decay function, Be$^{-R*d}$, of the TSA-biotin signal in which R is the decay constant and d is the distance to the central element of the kernel (B=5.86, *R*=3).

To correlate HFF LAD clusters with published FISH data (*Su et al., 2020*; *Figure 7C–D*), the x-y coordinates of FISH probes were first rotated using PCA such that the length (PC1) and width (PC2) of each nucleus aligns with its PC1 (X) and PC2 (Y) axes. The size of each nucleus was then normalized to fit in a 4x2×1 (X*Y*Z, arbitrary units) cuboid using a min-max normalization. Probes mapping within equatorial (EQ, Cluster 1) and non-equatorial (non-EQ, Cluster 2) LAD clusters were selected and overlayed to generate *Figure 7C*. Similarly, to correlate high-throughput FISH data with distance to centromeres, probes mapping to chromosome 4 (*Figure 7D*) were selected and their distances to centromeres were calculated using UCSC chromosome banding data (*Supplementary file 1*).

## Acknowledgements

This work was supported by the National Institutes of Health Common Fund 4D Nucleome Program grants U54DK107965 (ASB, BvS, DMG, and JM) and UM1HG011593 (JM, ASB, and DMG). Research at the Netherlands Cancer Institute is supported by an institutional grant of the Dutch Cancer Society and of the Dutch Ministry of Health, Welfare and Sports. The Oncode Institute is partially funded by the Dutch Cancer Society.

## Additional information

### Funding

| Funder | Grant reference number | Author |
|---|---|---|
| National Institutes of Health | U54DK107965 | David M Gilbert<br>Jian Ma<br>Bas van Steensel<br>Andrew S Belmont |
| National Institutes of Health | UM1HG011593 | David M Gilbert<br>Jian Ma<br>Andrew S Belmont |

The funders had no role in study design, data collection and interpretation, or the decision to submit the work for publication.

### Author contributions

Omid Gholamalamdari, Conceptualization, Writing – original draft, Writing – review and editing, Co-leader of biweekly online meetings of trainees to guide data analysis and hypothesis generation-Immunostaining and microscopy of nuclear locales (HCT116, K562); Software for segmentation and analysis of nuclear locale microscopy data; Data analysis subdividing LADs by nuclear locale positioning (Figure 6, Figure 6—figure supplement 1); Data analysis of positioning of genome relative to

equatorial plane and radial distances in Figure 7 and Figure 7—figure supplement 2; Comparisons of IMR90 imaging data and TSA-seq in Figure 7 and Figure 7—figure supplement 1; Model Summarizing Results; Ideas for and realization of model in Fig. 8; Writing of later drafts, Data curation, Formal analysis, Investigation, Methodology, Software, Supervision, Visualization; Tom van Schaik, Conceptualization, Writing – original draft, Writing – review and editing, Co-leader of biweekly online meetings of trainees to guide data analysis and hypothesis generation-DamID experiments; Data analysis correlating changes in gene expression and replication timing with nuclear locale changes in Figure 2, Figure 2—figure supplement 1; Analysis of p-w-v- and v-iLADs in Figure 4 and Figure 4—figure supplement 1; Analysis of genome positioning in response to LMNA and LBR DKO in Figure 5 and Figure 5—figure supplement 1; Data analysis of positioning of genome relative to equatorial plane and radial distances in Figure 7 and Figure 7—figure supplement 2, Data curation, Formal analysis, Investigation, Methodology, Software, Visualization; Yuchuan Wang, Conceptualization, Writing – original draft, Data curation and quality control; Analysis of Type 1 and II SON TSA-seq peaks in Figure 3 and Figure 3—figure supplement 1; SPIN analysis for Figure 7I, Data curation, Formal analysis, Visualization; Pradeep Kumar, Conceptualization, Immunostaining and microscopy of nuclear locales (H1, HFF and K562-LMNA/LBR DKO); TSA-seq for MKI67IP (all cell lines), LMNB1 (K562, H1 and HFF, K562 LMNA/LBR DKO), and SON (K562wt and K562 LMNA/LBR DKO); Development of initial idea for p-w-v fiLADS; Comparisons of IMR90 imaging data and TSA-seq in Figure 7 and Figure 7—figure supplement 1, Data curation; Liguo Zhang, Conceptualization, Writing – review and editing, TSA-seq for SON (K562, H1, HCT116, HFFc6) and LMNB1 (HCT116, H1, HFFc6); Analysis for Figure 2C and Figure 2—figure supplement 1D-E; Yang Zhang, Conceptualization, Writing – review and editing, Data curation and quality control; Software development and implementation for coordinated display and analysis of multi-dimensional scatterplots used to guide hypothesis generation in study; Analysis for Figure 3F, Data curation, Software; Gabriela A Hernandez Gonzalez, Writing – review and editing, Live-cell imaging data collection and analysis for Figure 3—figure supplement 1F; Athanasios E Vouzas, Early/late Repli-seq in K562-LMNA/LBR DKO cells; Peiyao A Zhao, Initial analysis for Figure 3F; David M Gilbert, Jian Ma, Bas van Steensel, Conceptualization, Resources, Supervision, Funding acquisition, Project administration, Writing – review and editing; Andrew S Belmont, Conceptualization, Resources, Supervision, Funding acquisition, Investigation, Writing – original draft, Project administration, Writing – review and editing, Development of initial idea for p-w-v fiLADS-Writing-later drafts

### Author ORCIDs
Omid Gholamalamdari https://orcid.org/0000-0002-5773-1205
Pradeep Kumar https://orcid.org/0000-0002-1060-1286
Jian Ma https://orcid.org/0000-0002-4202-5834
Bas van Steensel https://orcid.org/0000-0002-0284-0404
Andrew S Belmont https://orcid.org/0000-0002-6540-0801

Reviewer #3 (Public review): https://doi.org/10.7554/eLife.99116.4.sa1
Author response https://doi.org/10.7554/eLife.99116.4.sa2

---

## Additional files

### Supplementary files
Supplementary file 1. Genomic datasets that were generated as part of this study or reused from publicly available data or other published studies. This file contains five sheets listing TSA-seq, DamID seq, Repli-seq, ChIP-seq, and RNA-seq datasets.

MDAR checklist

### Data availability
Bed files corresponding to the genomic domains and datasets that we are uploading or previously have uploaded to 4DN can be found on the manuscript webpage on the 4DN website (https://data.4dnucleome.org/belmont_lab_nuclear_locale). The K562 LMNA, LBR, and LMNA/LBR DamID-seq, Repli-seq, and RNA-seq raw and processed data has been deposited to GEO (GSE263012). The locale imaging data and the corresponding segmentations can be found on Illinois Data Bank (https://doi.

org/10.13012/B2IDB-9792611_V1). The code used to generate figures in this manuscript, along with processed input and output files can be found have been reposited to the Illinois Data Bank (https://doi.org/10.13012/B2IDB-4383352_V1).

The following datasets were generated:

| Author(s) | Year | Dataset title | Dataset URL | Database and Identifier |
|---|---|---|---|---|
| Zhang L, Belmont AS | 2019 | Replicates of SON Ab2 TSA-seq version 2 Reaction Condition 2 on H1 cells | https://data.4dnucleome.org/experiment-set-replicates/4DNESC3D6NGQ/ | 4DNUCLEOME, 4DNESC3D6NGQ |
| Zhang L, Belmont AS | 2019 | Replicates of SON Ab2 TSA-seq version 2 Reaction Condition 2 on HCT116 cells | https://data.4dnucleome.org/experiment-set-replicates/4DNESH8LXMG3/ | 4DNUCLEOME, 4DNESH8LXMG3 |
| Kumar P, Belmont AS | 2021 | Replicates of MKI67IP TSA-seq Reaction Condition E on HFFc6 cells | https://data.4dnucleome.org/experiment-set-replicates/4DNESGAR9ZBW/ | 4DNUCLEOME, 4DNESGAR9ZBW |
| Zhang L, Belmont AS | 2019 | Replicates of SON Ab2 TSA-seq version 2 Reaction Condition 2 on HFFc6 cells | https://data.4dnucleome.org/experiment-set-replicates/4DNES85R9TIB/ | 4DNUCLEOME, 4DNES85R9TIB |
| Chen Y, Belmont AS | 2018 | Replicates of Lamin B TSA-seq v1 on K562 cells | https://data.4dnucleome.org/experiment-set-replicates/4DNESA1Z2ZVR/ | 4DNUCLEOME, 4DNESA1Z2ZVR |
| Zhang L, Belmont AS | 2019 | Replicates of SON Ab2 TSA-seq version 2 Reaction Condition 2 on K562 cells | https://data.4dnucleome.org/experiment-set-replicates/4DNESW3L596Q/ | 4DNUCLEOME, 4DNESW3L596Q |
| Kumar P, Belmont AS | 2023 | Replicates of MKI67IP TSA-seq Reaction Condition E (PBS 50% Sucrose, 1:300 tyramide biotin) 30 minute reaction on lmna-lbr-ko-k562 cells | https://data.4dnucleome.org/experiment-set-replicates/4DNESXHWK24G/ | 4DNUCLEOME, 4DNESXHWK24G |
| Kumar P, Belmont AS | 2023 | Replicates of Lamin B1 TSA-seq version 2 Reaction Condition 2 (PBS 50% Sucrose) Enhancement condition AI (1:10000 tyramide-biotin, 20 minute reaction) on lmna-lbr-ko-k562 cells | https://staging.4dnucleome.org/experiment-set-replicates/4DNESICMN7RP/#processed-files | 4DNUCLEOME, 4DNESICMN7RP |
| Kumar P, Belmont AS | 2023 | Replicates of SON Ab2 TSA-seq version 2 Reaction Condition 2 (PBS 50% Sucrose) Enhancement condition E (1:300 tyramide-biotin, 30 minute reaction) on lmna-lbr-ko-k562 cells | https://data.4dnucleome.org/experiment-set-replicates/4DNES2ZND18K/ | 4DNUCLEOME, 4DNES2ZND18K |
| van Schaik T, van Steensel B, Gilbert DM, Vouzas AE | 2024 | Genomic positioning in LBR and Lamin A K562 knockout cells | https://www.ncbi.nlm.nih.gov/geo/query/acc.cgi?acc=GSE263012 | NCBI Gene Expression Omnibus, GSE263012 |

*Continued on next page*

*Continued*

| Author(s) | Year | Dataset title | Dataset URL | Database and Identifier |
|---|---|---|---|---|
| Gholamalamdari O, Kumar P, Belmont A | 2024 | Nuclear locale immunofluorescence imaging and segmentation in four human cell lines | https://doi.org/10.13012/B2IDB-9792611_V1 | Illinois Data Bank, 10.13012/B2IDB-9792611_V1 |
| Gholamalamdari O, Belmont A | 2024 | Supporting material for Omid Gholamalamdari et al. 2024 | https://doi.org/10.13012/B2IDB-4383352_V1 | Illinois Data Bank, 10.13012/B2IDB-4383352_V1 |

The following previously published datasets were used:

| Author(s) | Year | Dataset title | Dataset URL | Database and Identifier |
|---|---|---|---|---|
| Gilbert D | 2021 | 16 fraction Repliseq and G1 fraction on HCT116 Tier 2 cells | https://data.4dnucleome.org/experiment-sets/4DNESNGZM5FG/ | 4DNCULEOME, 4DNESNGZM5FG |
| Gilbert D | 2019 | 16 fraction Repliseq and G1 fraction on H1-hESC Tier 1 cells | https://data.4dnucleome.org/experiment-sets/4DNESXRBILXJ/ | 4DNUCLEOME, 4DNESXRBILXJ |
| Gilbert D | 2017 | Early and Late 2 phase Repliseq on K562 Tier 2 cells | https://data.4dnucleome.org/experiment-sets/4DNESC2VFNP2/ | 4DNUCLEOME, 4DNESC2VFNP2 |
| Gilbert D | 2017 | Early and Late 2 phase Repliseq on HCT116 Tier 2 cells | https://data.4dnucleome.org/experiment-sets/4DNESXPNEE4Q/ | 4DNUCLEOME, 4DNESXPNEE4Q |
| Gilbert D | 2017 | Early and Late 2 phase Repliseq on H1-hESC Tier 1 cells | https://data.4dnucleome.org/experiment-sets/4DNESEJCRVNR/ | 4DNUCLEOME, 4DNESEJCRVNR |
| Gilbert D | 2018 | Early and Late 2 phase Repliseq on HFFc6 Tier 1 cells | https://data.4dnucleome.org/experiment-sets/4DNESIB37EU2/ | 4DNUCLEOME, 4DNESIB37EU2 |
| van Schaik T, van Steensel B | 2018 | LaminB1 DamID of K562 Tier 2 cells | https://data.4dnucleome.org/experiment-set-replicates/4DNESTAJJM3X/ | 4DNUCLEOME, 4DNESTAJJM3X |
| van Schaik T, van Steensel B | 2017 | LaminB1 DamID of HFFc6 Tier 1 cells | https://data.4dnucleome.org/experiment-set-replicates/4DNESXZ4FW4T/ | 4DNUCLEOME, 4DNESXZ4FW4T |
| van Schaik T, van Steensel B | 2018 | LaminB1 DamID of HCT116 Tier 2 cells | https://data.4dnucleome.org/experiment-set-replicates/4DNES24XA7U8/ | 4DNUCLEOME, 4DNES24XA7U8 |
| van Schaik T, van Steensel B | 2017 | LaminB1 DamID of H1-hESC Tier 1 cells | https://data.4dnucleome.org/experiment-set-replicates/4DNESXKBPZKQ/ | 4DNUCLEOME, 4DNESXKBPZKQ |

*Continued on next page*

*Continued*

| Author(s) | Year | Dataset title | Dataset URL | Database and Identifier |
|---|---|---|---|---|
| Kumar P, Belmont AS | 2021 | Replicates of MKI67IP TSA-seq Reaction Condition E on K562 cells | https://data.4dnucleome.org/experiment-set-replicates/4DNESI714SZN/ | 4DNUCLEOME, 4DNESI714SZN |
| Zhang L, Kumar P, Belmont AS | 2020 | Replicates of Lamin B1 TSA-seq v2 on HFFc6 cells | https://data.4dnucleome.org/experiment-set-replicates/4DNESMF4T7QQ/ | 4DNUCLEOME, 4DNESMF4T7QQ |
| Zhang L, Belmont AS | 2020 | Replicates of Lamin B1 TSA-seq v2 on HCT116 | https://data.4dnucleome.org/experiment-set-replicates/4DNESTE8NJOE/ | 4DNUCLEOME, 4DNESTE8NJOE |
| Kumar P, Belmont AS | 2021 | Replicates of MKI67IP TSA-seq Reaction Condition E on HCT116 | https://data.4dnucleome.org/experiment-set-replicates/4DNESAHA7E69/ | 4DNUCLEOME, 4DNESAHA7E69 |
| Zhang L, Kumar P, Belmont AS | 2020 | Replicates of Lamin B1 TSA-seq v2 on H1 | https://data.4dnucleome.org/experiment-set-replicates/4DNESGGXKI1H/ | 4DNUCLEOME, 4DNESGGXKI1H |
| Kumar P, Belmont AS | 2021 | Replicates of MKI67IP TSA-seq Reaction Condition E on H1 | https://data.4dnucleome.org/experiment-set-replicates/4DNESO6HFSAD/ | 4DNUCLEOME, 4DNESO6HFSAD |
| Alexander KA, Berger SL | 2021 | p53 transcription factor mediates nuclear speckle association of target genes | https://www.ncbi.nlm.nih.gov/geo/query/acc.cgi?acc=GSE154095 | NCBI Gene Expression Omnibus, GSE154095 |
| Wold B | 2020 | total RNA-seq in K562 | https://www.encodeproject.org/search/?searchTerm=ENCFF928NYA | ENCODE, ENCFF928NYA |
| Wold B | 2020 | total RNA-seq in HCT116 | https://www.encodeproject.org/search/?searchTerm=ENCFF435PHM | ENCODE, ENCFF435PHM |
| Wold B | 2020 | total RNA-seq in H1 | https://www.encodeproject.org/search/?searchTerm=ENCFF174OMR | ENCODE, ENCFF174OMR |
| Yue F | 2019 | RNA-seq on HFFc6 (Tier 1) cells | https://data.4dnucleome.org/files-processed/4DNFI5MR6C3G/ | 4DNUCLEOME, 4DNFI5MR6C3G |

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
