## [Editor Report · eLife Assessment]

In this **valuable** study, the authors integrate several datasets to describe how the genome interacts with nuclear bodies across distinct cell types and in Lamin A and LBR knockout cells. They provide **convincing** evidence to support their claims and particularly find that specific genomic regions segregate relative to the equatorial plane of the cell when considering their interaction with various nuclear bodies.

---

## [Referee Report · Reviewer #3 (Public review)]

Summary:

Golamalamdari, van Schaik, Wang, Kumar Zhang, Zhang and colleagues study interactions between the speckle, nucleolus and lamina in multiple cell types (K562, H1, HCT116 and HFF). Their datasets define how interactions between the genome and the different nuclear landmarks relate to each other and change across cell types. They also identify how these relationships change in K562 cells in which LBR and LMNA are knocked out.

Strengths:

Overall, there are a number of datasets that are provided, and several "integrative" analyses performed. This is a major strength of the paper, and I imagine the datasets will be of use to the community to further probed and the relationships elucidated here further studied. An especially interesting result was that specific genomic regions (relative to their association with the speckle, lamina, and other molecular characteristics) segregate relative to the equatorial plane of the cell.

Weaknesses:

The experiments are primarily descriptive, and the cause-and-effect relationships are limited (though the authors do study the role of LMNA/LBR knockdown with their technologies).

Comments on revisions:

I have no additional comments. I appreciate the authors responding to my previous comments. I anticipate the datasets and concepts raised will be helpful to many investigators in the field.

---

## [Author Response]

The following is the authors’ response to the previous reviews

**Response to Public Reviews:**

We would like to thank the reviewers and editors once more for their time and effort in reviewing our manuscript. Below we discuss specifically our response to the recommendations of Reviewer 2, which were the only substantial changes we made to the manuscript.

**Reviewer 2 recommendation:**
"My only remaining suggestion is that the authors acknowledge and cite the work of other groups which have similarly found different subsets of LADs based on various molecular/epigenetic features:(1) doi.org/10.1101/2024.12.20.629719(2) PMID: 25995381(3) PMID: 36691074(4) PMID: 23124521 (fLADs versus cLADs, as described by the authors themselves) The exact subtypes of LADs might be different based on the features examined, but others have found/implicated the existence of different types of LADs. Hence, the pwv-LAD should be contextualized within these findings (which they do relative to v-fiLADs)."

We thank the reviewer for this suggestion and for these references. We think that the best place to go into depth about how our work relates to these references would be in an appropriate review article.

However, we did read these references carefully and responded, as described below, by adding additional clarifying text in the manuscript as well as mention of articles specifically relevant to our description of our results.

(1) Reviewer 2 wrote specifically, "Hence, the pwv-LAD should be contextualized within these findings (which they do relative to v-fiLADs)"

We are not sure exactly what Reviewer 2 means here. In this manuscript we defined p-w-v iLADs, not LADs. So, it would be inappropriate to compare a subset of iLAD regions with different types of LADs.

If this was the meaning of Reviewer 2, then other readers might have similar confusion. Therefore, we added the following clarifying text in red:

"Several previous studies have used varying approaches to subdivide LADs further into distinct subsets of LADs with different biochemical and/or functional properties (Martin et al., 2024; Meuleman et al., 2013; Shah et al., 2023; Zheng et al., 2015). However, in this Section we focused instead on asking whether regions specifically within iLADs might show differential localization relative to the lamina and/or nucleoli and, if so, whether these regions would show different levels of gene expression. More specifically, analogously to how gene expression hot-zones appeared as local maxima in speckle TSA-seq with early DNA replication timing, we asked whether iLAD regions that appeared as local maxima in lamina proximity mapping signals would correspond to iLAD regions with locally reduced gene expression levels and later DNA replication timing relative to their flanking iLAD sequences. Our rationale was that these iLAD regions might represent chromatin domains that together with their flanking iLAD regions would typically localize well within the nuclear interior but in a fraction of the cell population would loop back and attach at the nuclear periphery."

(2) We also added the following text near the end of the section about p-w-v iLADs to place them in the context of one class of "LADs" identified by ChIP-seq rather than DamID. We use quotation marks since the approach used produced a segmentation that included a nearly 50/50 mix of iLAD and LAD regions, as identified by DamID, for this class of domains.

"We note that in a previous study a three-state Hidden Markov Model (HMM) segmented lamin B ChIP-seq data into two chromatin domain states with extensive overlap with LADs defined by lamina DamID (Shah et al., 2023). Whereas the late replicating, low gene density/expression "T1 LAD" state showed very high overlap (98%) with LADs defined by DamID, the intermediate replicating, intermediate gene expression "T2 LAD" state showed only 47% overlap with LADs defined by DamID. This was partly a result of the HMM segmentation algorithm but also due to substantial differences between the lamina ChIPseq versus DamID signals for reasons that remain unclear. The subset of p-w-v iLADs included in T2 comprise only a small percentage of the total T2 LAD coverage, which includes both other iLAD and LAD regions. Thus, the p-w-v iLADs we identified here represent a novel and distinct class of iLAD chromatin domains, not previously described."

(3) Alternatively, what Reviewer 2 might be suggesting implicitly is that we should start with the regions identified as p-w-v iLADs in one cell type and then identify all of those p-w-v iLADs which instead exist as LADs in a second cell type. Once we have identified their LAD equivalents in a second cell type we could then ask whether they possess special characteristics such that they correspond to a specific type of LAD subset. Finally, we could then ask how that specific type of LAD subset compared to the different subtypes of LADs identified by other groups and, in particular, the references Reviewer 2 provided.

We agree that would be an interesting future direction, but we consider that as outside the scope of this current manuscript. We note that we did no such analysis of the characteristics of LADs which existed as p-w-v iLADs in a different cell line. We save that for a possible future analysis, ideally in the same cell types as used in the cited references to allow a more direct comparison.

(4) Finally, we added text in the Discussion that relates our analysis of the differential SON and LMNB1 TSA-seq signals for different LAD regions, and how these correlate with different histone modifications, with results from the recent preprint cited by Reviewer 2. Note that we could not directly correlate our results from human cells with the three classes of LADs described in MEFs by this preprint.

"Fourth, we show how LAD regions showing different histone marks- either enriched in H3K9me3, H3K9me2 plus H2A.Z, H3K27me3, or none of these marks- can differentially segregate within nuclei. These results support the previous suggestion of different "flavors" of LAD regions, based on the sensitivity of the autonomous targeting of BAC transgenes to the lamina to different histone methyltransferases (Bian et al., 2013). Differential nuclear localization also was recently inferred by the appearance of different Hi-C Bsubcompartments, which similarly were differentially enriched in either H3K9m3, H3K27me3, or the combination of H3K9me2 and H2A.Z (Spracklin et al., 2023). More recently, and while this paper was in revision, a new study described segmenting mouse embryonic fibroblast LADs into three clusters using histone modification profiling (Martin et al., 2024). Interestingly, these three LAD clusters also most notably differed by their dominant enrichment of either H3K9me3, H3K9me2, or H3K27me3. Thus, three orthogonal approaches have converged on identifying different LAD regions showing differential enrichment either of H3K9me3, H3K9me2, or H3K27me3. Here, our use of TSA-seq directly measures and assigns the intranuclear localization of these different LAD regions to different nuclear locales."